# HIV-1 exploits LBPA-dependent intraepithelial trafficking for productive infection of human intestinal mucosa

Anusca G. Rader[1,2,3], Alexandra P. M. Cloherty[1,2], Kharishma S. Patel[1,2], Dima D. A. Almandawi[1,2], Dasja Pajkrt[4], Katja C. Wolthers[5], Adithya Sridhar[4,5], Sterre van Piggelen[1,2], Liselotte E. Baaij[1,2], Renée R. C. E. Schreurs[1,2,3], Carla M. S. Ribeiro[1,2,3]*

1 Amsterdam UMC, location University of Amsterdam, Experimental Immunology, Amsterdam, The Netherlands, 2 Amsterdam institute for Immunology & Infectious Diseases, Amsterdam, The Netherlands, 3 Amsterdam Gastroenterology, Endocrinology & Metabolism, Amsterdam, The Netherlands, 4 Amsterdam UMC, location University of Amsterdam, Pediatric Infectious Diseases, Emma Children's Hospital, Amsterdam, The Netherlands, 5 Amsterdam UMC, location University of Amsterdam, Medical Microbiology, Amsterdam, The Netherlands

* c.m.ribeiro@amsterdamumc.nl

**Data Availability Statement:** All relevant data are within the manuscript and its Supporting Information files.

**Funding:** The researcher AGR and APMC were supported by an Amsterdam UMC grant and an

## Abstract

The gastrointestinal tract is a prominent portal of entry for HIV-1 during sexual or perinatal transmission, as well as a major site of HIV-1 persistence and replication. Elucidation of underlying mechanisms of intestinal HIV-1 infection are thus needed for the advancement of HIV-1 curative therapies. Here, we present a human 2D intestinal immuno-organoid system to model HIV-1 disease that recapitulates tissue compartmentalization and epithelial-immune cellular interactions. Our data demonstrate that apical exposure of intestinal epithelium to HIV-1 results in viral internalization, with subsequent basolateral shedding of replication-competent viruses, in a manner that is impervious to antiretroviral treatment. Incorporation of subepithelial dendritic cells resulted in HIV-1 luminal sampling and amplification of residual viral replication of lab-adapted and transmitted-founder (T/F) HIV-1 variants. Markedly, intraepithelial viral capture ensued an altered distribution of specialized endosomal pathways alongside durable sequestration of infectious HIV-1 within lysobisphosphatidic acid (LPBA)-rich vesicles. Therapeutic neutralization of LBPA-dependent trafficking limited productive HIV-1 infection, and thereby demonstrated the pivotal role of intraepithelial multivesicular endosomes as niches for virulent HIV-1 within the intestinal mucosa. Our study showcases the application of primary human 2D immune-competent organoid cultures in uncovering mechanisms of intestinal HIV-1 disease as well as a platform for preclinical antiviral drug discovery.

## Author summary

The human intestine forms an important roadblock against viruses, including HIV-1. Nevertheless, upon sexual contact HIV-1 is able to penetrate the intestinal barrier and infect cells of our immune system within the intestine, resulting in HIV-1 infection of the

Amsterdam UMC PhD scholarship, respectively. RRCE and CMSR received funding from the Stichting Steun Emma Kinderziekenhuis under grant agreement WAR2021-18. DP, KCW, AS and CMSR acknowledge research support from the PPP Allowance (Focus-on-Virus) by Health Holland, Top Sector Life Sciences & Health. CMSR has received funding for this study from the Dutch Research Council (NWO-ZonMw) through VIDI and ASPASIA grants under grant agreements 91718331 and 015.014.030, respectively. The funders had no role in study design, data collection and analysis, decision to publish, or preparation of the manuscript.

**Competing interests:** The authors have declared that no competing interests exist.

human body. Little is still known about how HIV-1 can outsmart this protective intestinal barrier and establish an infection. Here, we reconstructed the human intestine outside the human body and developed a novel mini-gut model containing relevant immune cells that allowed us to track the HIV-1's footsteps throughout the intestine in the laboratory. Using advanced imaging technology, we have uncovered that HIV-1 specifically hijacks vesicles studded with the particular molecule (LBPA) that are produced inside of the intestinal barrier cells to subsequently spread infection to adjacent intestinal cells. Importantly, blockade of these LBPA-enriched vesicles with an antibody suppressed intestinal HIV-1 infection. Our study substantiates the use of upgraded human gut models to deepen our knowledge about human viral diseases as well as underscores the development of therapies targeting LBPA for superior control of intestinal HIV-1 disease in people living with HIV-1.

## Introduction

Despite considerable efforts there is still no available cure for human immunodeficiency virus type 1 (HIV-1) infection. HIV-1 therefore remains a major public health threat, with 1,3 million new infections and 630,000 deaths in 2022 alone [1]. Transmission via the intestinal mucosa has been estimated to be responsible for approximately one quarter of these HIV-1 infections worldwide with selective transmission of CCR5-using over CXCR4-using HIV-1 variants [2–5]. In addition, compared to other human mucosal tissues, the intestinal mucosa houses the highest concentration of HIV-1 mucosal target cells, including dendritic cells (DCs), macrophages, and CD4+ T cells. The intestinal mucosa therefore represents a major anatomical site in which HIV-1 can replicate, disseminate, and latently persist in reservoirs [6].

In mucosal tissues, DCs are amongst the first immune target cell to encounter HIV-1 [7,8]. Due to their functionality as antigen presenting cells, immature DCs residing in the mucosa can capture and internalize virions as well as support HIV-1 replication [9–13]. In concordance, seminal studies have shown that HIV-1 exploits this DC migratory capacity, which elicits HIV-1 transmission to other target cells [14–17]. Intestinal epithelial cells on the contrary do not express the primary receptor for HIV-1, CD4 receptor [18,19], and are therefore not productively infected with the virus. Nonetheless, the intestinal epithelium itself has also been implicated in mucosal HIV-1 pathogenicity [6,19,20]. Due to the lack of relevant models for the dissection of early mucosal events during HIV-1 infection, including the routes utilized by HIV-1 to migrate through intestinal epithelium towards subepithelial mucosal immune cells, such as DCs, remains poorly elucidated.

To address this question, it is essential to employ a human-relevant model. Although cell lines and animal models have been widely used to model human diseases, cell lines have mutations or genetic alterations differing from primary tissue, resulting in the increased chance of atypical cellular responses to viral exposure [21]. In addition, our previous research has demonstrated that intracellular immune defenses against HIV-1 are both species- and cell-specific [11,22]. Technological advances have enabled the generation of *in vitro* organoid models derived from human stem cells. Organoids are self-organized 3D tissue models, which recapitulate the key structural and biological complexity of the *in vivo* tissue, and have been shown to be applicable in a wide variety of research fields [23–25]. Recently, a standardized 2D primary intestinal epithelium model, derived from human 3D organoids with a similar differentiation profile to the human intestine, was developed for the study of host-pathogen interactions [26]. We have further substantiated its applicability for studying mechanisms of enteric viral entry [27].

Here, we present a human intestinal immuno-organoid to model intestinal HIV-1 disease. We take advantage of the well-characterized polarized intestinal epithelium model and add further complexity by the addition of human DCs. This multicellular organoid model thus more closely mimics the intercellular interactions and immune compartmentalization of the human intestinal mucosa *in vivo*. The establishment of the epithelium-DC co-culture model enabled us to study intraepithelial HIV-1 sequestration and identify a vesicle-mediated infection mechanism within the intestinal mucosa, which is blockable by pharmaceutical intervention. In addition, we reconstructed *in vitro* mucosal HIV-1 targeting and subsequent propagation of lab-adapted and clinically-relevant HIV-1 variants by lumen-sampling sube-pithelial DCs. Thereby, our model provides a human 2D immune-competent organoid platform, amenable for experimental manipulation and in-depth studies on viral infection mechanisms *in vitro*, which is also well-suited for antiviral drug screening.

## Results

### Primary intestinal epithelial cells shed infectious HIV-1 in *in vitro* reconstructed gut epithelium

During sexual contact or perinatal transmission, HIV-1 must traverse epithelial barriers at mucosal surfaces to gain access to underlying cellular targets and establish mucosal HIV-1 infection [28–31]. The intestinal mucosa is thereby a major site for HIV-1 infection and pathogenesis [2,6]. Accumulating evidence supports that, whilst not targets for viral replication, intestinal epithelial cells play a cardinal role in the establishment of systemic HIV-1 infection [19,20,32]. Herein, we set out to model intestinal HIV-1 transmission *in vitro* by using 2D human intestinal organotypic cultures to uncover mechanisms of mucosal HIV-1 invasion.

To this end, we utilized a validated primary human fetal intestinal epithelial (IE) culture system, in which intestinal epithelial cells are grown on cell culture inserts, and which recapitulates the major characteristics of intestinal epithelium *in vivo* [Fig 1A] [26,27]. Generation of a polarized epithelial monolayer was confirmed by confocal imaging [Fig 1B]. Epithelial monolayer integrity was assessed by trans-epithelial electrical resistance (TEER) longitudinal measurements and paracellular permeability from apical to basolateral compartments was appraised using fluorescein isothiocyanate-conjugated dextran (4KDa; FD4) permeation rate [26,27]. Both TEER and FD4 permeation rates indicated the formation of an epithelial monolayer with tight barrier function prior to virus exposure on day 14 [Fig 1A, 1C and 1D]. We next aimed to assess HIV-1 transmission across the epithelium. To this end, confluent and non-permeable IE cultures were subsequently inoculated with HIV-1 at the apical side for 72h, after which basolateral medium was harvested and co-cultured with the HIV-1 permissive U87.CD4.CCR5 cell line [Fig 1A]. Across multiple donors, incubation with epithelial cell-derived basolateral medium resulted in the productive HIV-1 infection of U87.CD4.CCR5 target cells [Fig 1E and 1F]. These data indicate that exposure of intact polarized intestinal epithelium to HIV-1 is ensued by viral transport through the epithelial barrier, leading to basolateral release of infectious HIV-1 particles with subsequent viral spread to target cells.

### HIV-1 transmission by *ex vivo* intestinal-derived epithelial cells is impervious to prophylactic antiretroviral therapy

Current HIV-1 preventative or suppressive treatments consist of direct-acting antivirals that intervene in HIV-1 replication and are suitable in suppressing productive HIV-1 infection of CD4+ target cells [33]. We isolated *ex vivo* fetal intestinal cells, both epithelial and lamina propria-derived immune cells, and subjected them to pretreatment with antiretrovirals prior to

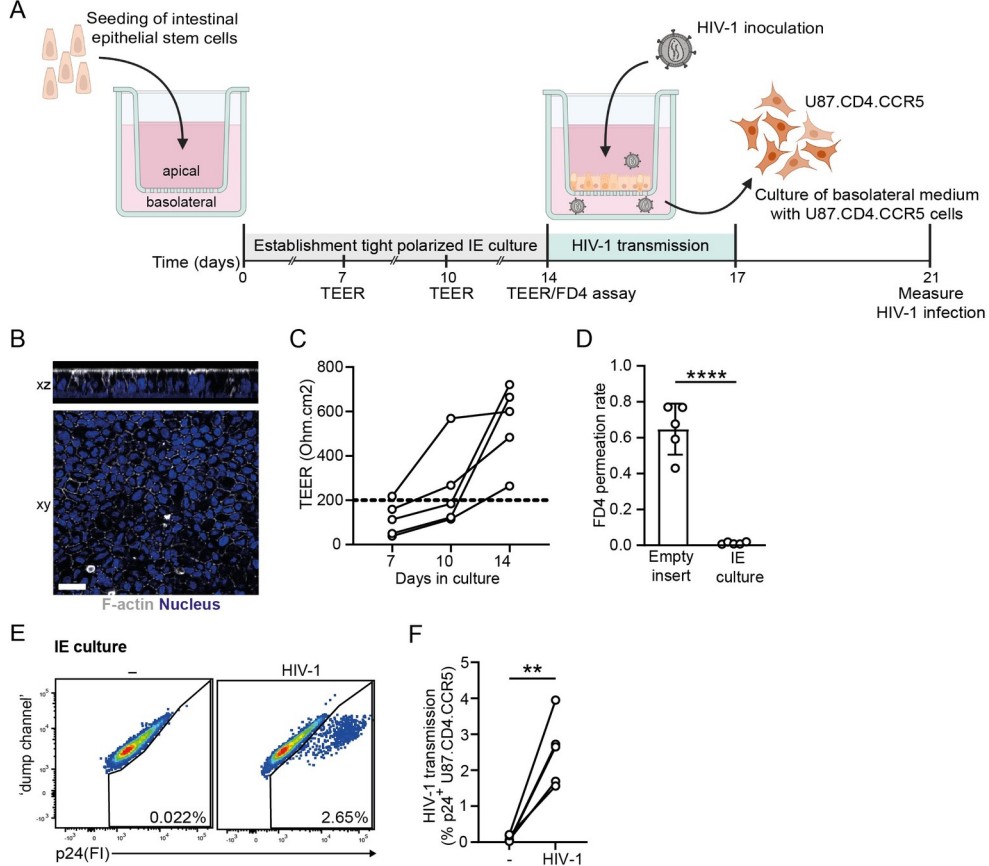

**Fig 1. Human primary intestinal epithelium transmit infectious HIV-1.** (**A**) Schematic representation of the establishment of confluent and non-permeable intestinal epithelium (IE) culture model using standardized trans-epithelial electrical resistance (TEER) and FITC-dextran paracellular quantitative assays, respectively, to study HIV-1 infection of human intestine *in vitro*. (**B**) Representative confocal images of top view (xz) and side view (xy) of the IE cultures resting on a 3.0 micron pore 24-well cell culture insert. F-actin (phalloidin) is shown in grey and nuclei (DAPI) in blue. Scale bar = 20 micron. Representative of *n* = 2 donors. (**C**) Integrity of IE cultures, determined by longitudinal TEER measurements in IE cultures at days 7, 10, and 14. Open circles represent individual gut donors, *n* = 5. TEER value ≥200 Ohm.cm2 (dashed line) indicative of a confluent monolayer prior to HIV-1 inoculation. (**D**) Paracellular permeability of IE cultures at day 14, determined by 4 kDa FITC-conjugated dextran (FD4) permeation rate of monolayers 4h post-FD4 addition. Permeability is expressed as FD4 permeation rate: FD4 basolateral$_{t = 4}$(μg)/FD4 apical$_{t = 0}$(μg), empty inserts included as negative control. Data are mean ± SD of *n* = 5 donors, Student's paired *t*-test, **** $p < 0.0001$. (**E,F**) HIV-1 transmission by IE cultures, determined by intracellular p24 staining of U87.CD4.CCR5 cell line by flow cytometry analyses. IE cultures were apically exposed to HIV-1 NL4.3BaL for 72h. Subsequently, basolateral supernatant was collected and co-cultured with HIV-1 permissive U87.CD4.CCR5 target cells for 96h. (**E**) Representative flow cytometry plots and (**F**) quantification. *n* = 5 donors, Student's paired *t*-test, ** $p < 0.01$.

HIV-1 inoculation [Fig 2A]. Indeed, fetal intestinal lamina propria-derived cells, containing the majority of the subepithelial HIV-1 permissive cells [12,34], readily induced HIV-1 transmission *ex vivo* to target U87.CD4.CCR5 cells, which was suppressed by prophylactic treatment with nucleotide analog reverse transcriptase inhibitor Tenofovir (TFV) over a wide concentration range [Fig 2B and 2C]. In contrast, fetal intestinal epithelium-mediated HIV-1 transmission to target cells was unaltered by TFV pre-treatment [Fig 2D and 2E]. Moreover, prophylactic treatment with CCR5 antagonist Maraviroc (MCV) or nucleoside analog reverse transcriptase inhibitor Emtricitabine (FTC), and to a lesser extent protease inhibitor Indinavir (IDV), suppressed HIV-1 transmission by lamina propria-derived cells [Fig 2F], but not the

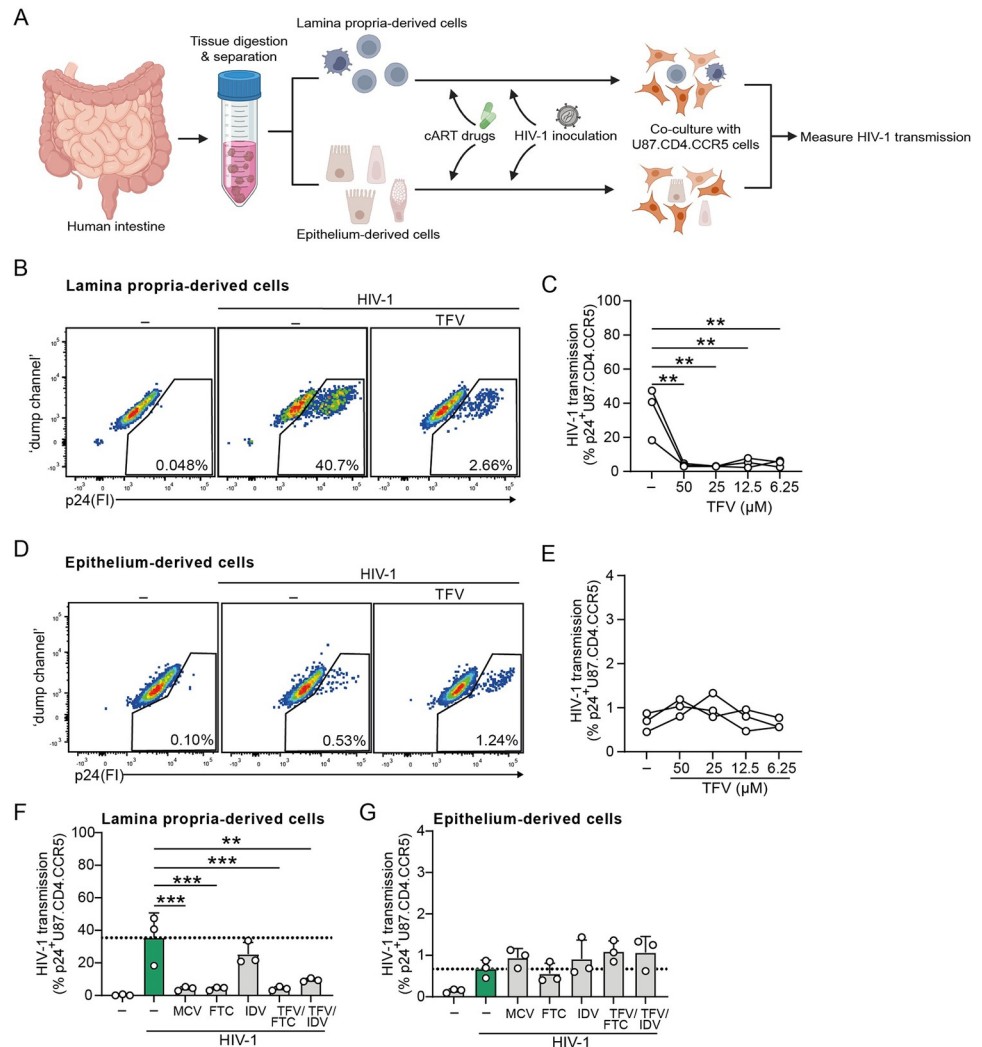

**Fig 2. Prophylactic antiretroviral treatment does not suppress HIV-1 transmission by intestinal epithelial cells *ex vivo*.** (**A**). Schematic representation of lamina propria-derived cells or epithelium-derived cells harvested from *ex vivo* human intestine tissue to study the effect of direct-acting antiretrovirals on HIV-1 transmission. (**B-G**) HIV-1 transmission by primary intestinal lamina propria-derived or epithelium-derived cells, determined by intracellular p24 staining of U87.CD4.CCR5 cell line by flow cytometry analyses. (**B,C**) Lamina propria-derived cells or (**D,E**) epithelium-derived CD45⁻ sorted cells were pre-treated with nucleotide reverse-transcriptase inhibitor Tenofovir (TFV, 50, 25, 12.5 or 6.25 µM) for 2h or left untreated prior to 48h exposure to HIV-1 NL4.3BaL. (**F**) Lamina propria-derived cells or (**G**) epithelium-derived CD45⁻ sorted cells were pre-treated with CCR5-antagonist Maraviroc (MCV, 30 µM), nucleoside reverse-transcriptase inhibitor Emtricitabine (FTC, 1 µM), protease inhibitor Indinavir (IDV,1 µM), or a combination of TFV (25 µM) with either FTC (0.2 µM) or IDV (0.2 µM) for 2h or left untreated prior to exposure to HIV-1 NL4.3BaL for 48h. (**B-G**) Subsequently, cells were washed extensively to remove input virus, and then co-cultured with permissive U87.CD4.CCR5 cells for 72h. Representative flow cytometry plots of HIV-1 transmission by TFV-treated lamina propria-derived cells (**B**) or TFV-treated epithelium-derived cells (**D**), and quantification (**C,E,F,G**). *n* = 3 donors, One-way ANOVA, $**p<0.01$, $***p<0.001$. (**F,G**) Data are mean ± SD.

epithelial-mediated HIV-1 transmission to U87.CD4.CCR5 cells [Fig 2G]. The reduced suppressive effect of IDV treatment on HIV-1 transmission by lamina-propria derived cells (Fig 2F) and direct infection of U87.CD4.CCR5 cells [S1 Fig], as assessed by intracellular HIV-1 capsid p24 protein expression, is attributed to IDV's inhibition of extracellular release of mature HIV-1 virions rather than directly interfering on earlier steps of HIV-1 replication cycle, such as translation of viral capsid protein [35]. In line with commonly prescribed two-

drug cART regimens [36–39], the combinatory treatment of TFV with either FTC or IDV prevented *cis*-infection of HIV-1 by lamina propria-derived cells [Fig 2F]. In contrast, HIV-1 transmission by epithelial-derived cells remained unaffected by these two-drug combinatory treatments [Fig 2G]. These results underline the ability of epithelial cells to promote intestinal HIV-1 *trans*-infection while remaining invulnerable to antiretroviral therapeutics, and highlight the role of subepithelial mucosa-associated immune cells in amplifying local HIV-1 replication upon intestinal HIV-1 invasion.

## Human primary 2D intestinal immuno-organoid model recapitulates intestinal mucosal HIV-1 transmission *in vitro*

Seminal reports on intestinal explant cultures have pointed to DCs as predominant cellular targets early after mucosal HIV-1 entry [32,40]. The *in vivo* intestinal mucosa consists of an epithelial layer forming the luminal surface of the intestine. Underneath is the lamina propria, a thin layer of connective tissue where most of the intestinal immune cells, including DCs, reside [41]. To mimic the intestinal architecture and to investigate the cellular mechanisms underlying mucosal HIV-1 transmission, herein a human intestinal infection organoid model was reconstructed *in vitro*. Human intestinal epithelium-DC (IEDC) co-cultures were established via sequential seeding of primary intestinal epithelial stem cells followed by human primary DCs onto a permeable membrane of 3.0 micron (cell culture inserts). First, $1*10^5$ intestinal epithelial stem cells were seeded onto the collagen-coated apical compartment of the cell culture insert, and cultured in validated media formulations [26] to form a tight and differentiated epithelial barrier [Figs 1A–1D and 3A]. Next, the insert was inverted and an optimized concentration of prototypical $2*10^5$ immature monocyte-derived DCs, with validated phenotypic profiling, cell viability and susceptibility to productive HIV-1 infection [S2A–S2G Fig], was seeded per insert onto the basolateral side of the insert membrane for 2 hours to allow cell attachment [Fig 3A]. Co-cultures were subsequently allowed to equilibrate for an additional day within organoid media supplemented with GM-CSF, prior to virus inoculation [Figs 3A and S2F]. TEER measurements in combination with confocal imaging confirmed the preservation of barrier integrity in the IEDC co-culture model, with viable DCs underlying the polarized intestinal epithelium [Fig 3A and 3B]. Furthermore, z-axial sectioning demonstrates the successful attachment of DCs to the basolateral compartment as well as preservation of DC sampling functionality as demonstrated by the formation of dendritic protrusions through the membrane and across the apical compartment 24h post co-culture [Fig 3C]. This human primary co-culture system thus incorporates an immune cell compartment whilst mimicking tissue compartmentalization and intercellular interactions.

Next, we assessed the pertinence of the 2D human intestinal immuno-organoids as a quintessential *in vitro* model of intestinal HIV-1 infection. Cross-sectional and 3D rendered visualization demonstrated the co-localization of HIV-1 p24 capsid with DC marker CD11c at 72h post infection, thereby indicating luminal HIV-1 sampling by DCs across the intestinal epithelium [Fig 3D and 3E]. To further assess DC activation and infection, we subjected the basolateral compartment of virus-exposed IEDC co-cultures to enzymatic cell detachment solution, which enabled the retrieval of the CD45$^+$ immune fraction composed of CD11C$^+$DC-SIGN$^+$DCs accompanied by a subpopulation of CD45$^-$ cells (epithelial fraction) [Fig 3F]. Upon HIV-1 infection, the basolateral detached DCs exhibit a semi-mature phenotype as evident from upregulation of activation marker CD83 and to a lesser degree CD86 [Fig 3G]. Flow cytometer analyses further supported the prominent HIV-1 infection of basolateral detached DCs in contrast to the co-detached CD11c$^-$ non-DC fraction [Figs 3H, 3I and S3A]. Furthermore, subsequent co-culture of basolateral detached DCs derived from HIV-exposed

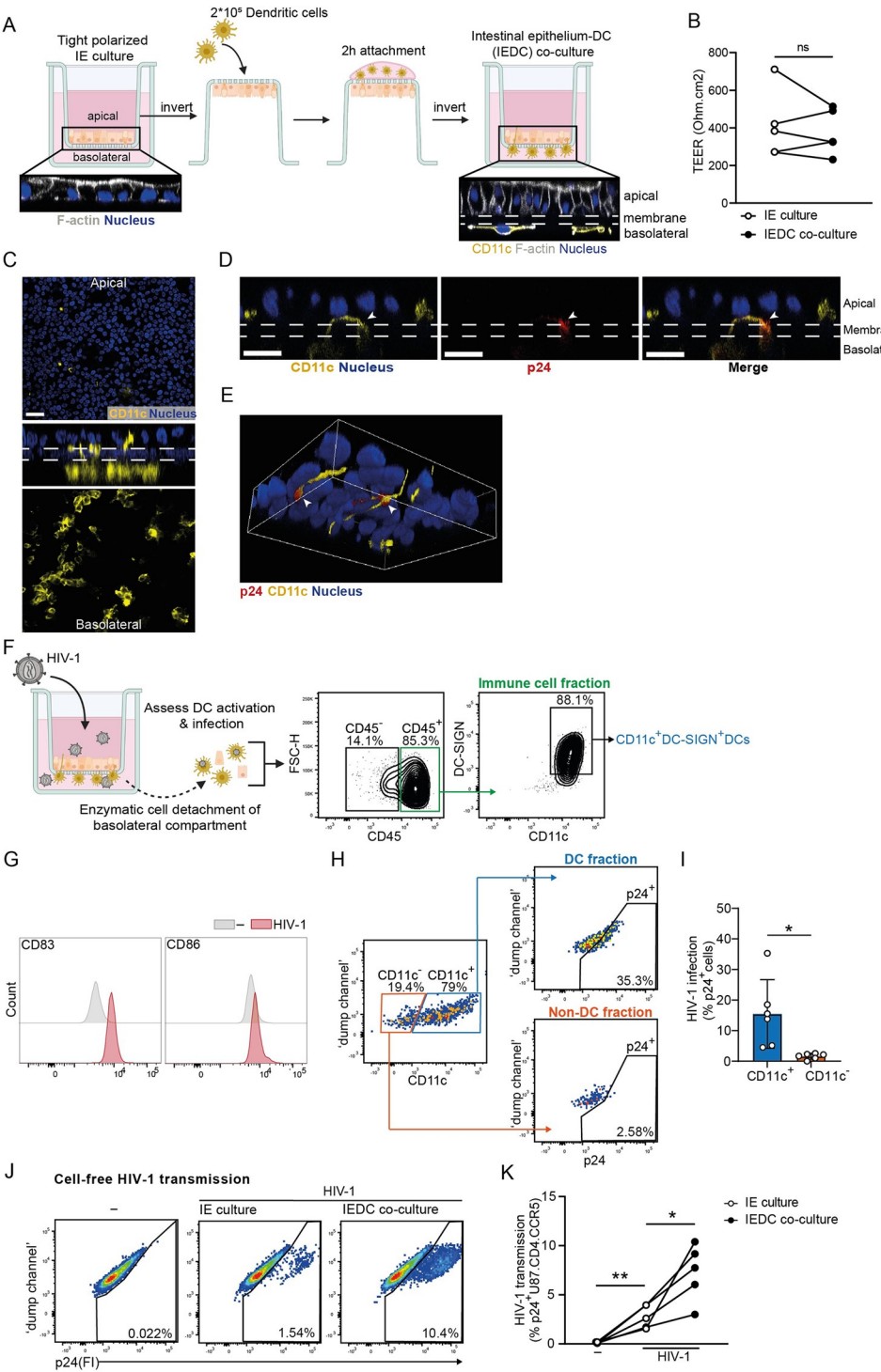

**Fig 3. Co-culture of human intestinal epithelium with dendritic cells results in enhanced HIV-1 transmission. (A)**
Schematic representation of the experimental protocol to generate intestinal epithelium-DC (IEDC) co-culture model.
Confocal microscopy images are z-stacks of IE culture and IEDC co-culture. DC surface marker CD11c (yellow), F-
actin (grey), and nuclei (blue), insert membrane (white dashed line). (**B**) Integrity of intestinal epithelial monolayers,
determined by TEER measurements pre- (open circles) and post- (closed circles) co-culture with DCs, *n* = 4 donors,
Student's paired *t*-test, ns = non-significant. (**C**) Visualization of IEDC co-culture model by confocal microscopy
analyses at the apical (top panel) or basolateral (bottom panel) side, and cross-sectional view (z-stack, middle panel).
DC surface marker CD11c (yellow) and nuclei (blue). Dashed lines indicate the separation of apical and basolateral

compartments by the insert membrane. Scale bar = 20 micron, representative of $n$ = 3 donors. (**D**) Cross-sectional view (z-stack) and (**E**) 3D rendered visualization of colocalization of DC protrusions with viral capsid in IEDC co-culture model (white arrowheads), upon apical exposure to HIV-1 NL4.3BaL for 72h. DC surface marker CD11c (yellow), HIV-1 p24 capsid (red) and nuclei (blue). Scale bar = 20 micron, representative of $n$ = 2 donors. (**F**) Schematic representation of enzymatic (Accutase)-based treatment of basolateral compartment from IEDC co-cultures, representative flow cytometry plots depict the immune cell fraction (CD45$^+$), non-immune epithelial cell fraction (CD45$^-$) and DC fraction (CD45$^+$CD11c$^+$DC-SIGN$^+$) detached from IEDC co-cultures, $n$ = 3 donors, for downstream analyses of (**G**) DC activation and (**H**) HIV-1 infection of DCs. (**G,H**) IEDC co-cultures were apically exposed to HIV-1 NL4.3BaL for 72h followed by enzymatic treatment of the basolateral compartment to retrieve DCs. (**G**) Representative histograms showing the expression of canonical activation markers CD83 and CD86 of CD45$^+$CD11c$^+$DC-SIGN$^+$ DCs upon HIV-1 infection as compared to uninfected, $n$ = 3 donors. (**H,I**) HIV-1 infection of basolateral DC fraction (CD11c$^+$p24$^+$) and basolateral non-DC fraction (CD11c$^-$p24$^+$), determined by flow cytometer. Data are mean ± SD of $n$ = 6 donors. (**J,K**) HIV-1 transmission by IE cultures and IEDC co-cultures, determined by intracellular p24 staining of U87.CD4.CCR5 cell line by flow cytometry analyses. IE cultures (open circles) or cognate IEDC co-cultures (closed circles) were apically exposed to HIV-1 NL4.3BaL for 72h. Subsequently, basolateral supernatants were collected and co-cultured with HIV-1 permissive U87.CD4.CCR5 target cells for 96h. (**J**) Representative flow cytometry plots and (**K**) quantification. $n$ = 5 donors, Student's paired $t$-test, $^*p<0.05$, $^{**}p<0.01$.

IEDC with target cells resulted in HIV-1 transmission to U87.CD4.CCR5 cells through cell-cell contact [S3B–S3D Fig], which underscores the productive HIV-1 infection of basolateral DCs upon apical viral exposure.

Next, to assess the cell-free HIV-1 transmission across IEDC co-cultures, the basolateral medium from HIV-1 exposed IEDC co-cultures was harvested and co-cultured with the permissive U87.CD4.CCR5 cell line. Notably, viral exposure of IEDC co-cultures, as compared to parallel HIV-1 exposed IE cultures without DC co-culture, resulted in consistently higher rates of HIV-1 transmission [Fig 3J and 3K]. The increased infection of U87.CD4.CCR5 cells in the IEDC co-cultures compared to IE culture reflects the enhanced basolateral release of replicative virus in the co-culture setting, underscoring the critical role of epithelial-DC interactions in amplifying mucosal viral production. Additionally, we show that IEDC co-cultures support the transmission of CCR5-tropic lab-adapted (NL4.3BaL) as well as transmitted/founder HIV-1 strains (CHO58, THRO) above that of CXCR4-using NL4.3 [Fig 4], which recapitulates the *in vivo* selective transmission of R5 HIV-1 [5,32,42]. Altogether, these data demonstrate that this newly established *in vitro* 2D human primary immuno-organoid model simulates early *in vivo* mucosal events of intestinal HIV-1 invasion, and underline the role of epithelial-DC interactions at the origin of mucosal HIV-1 propagation.

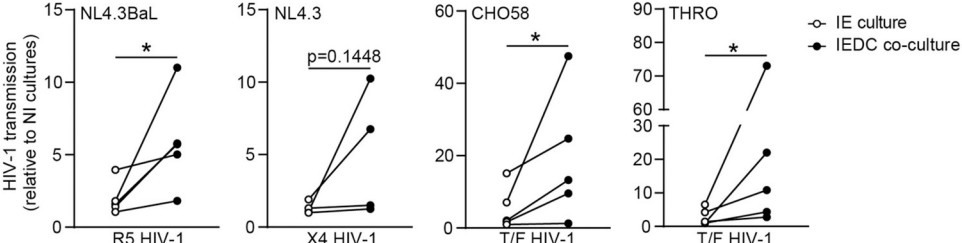

**Fig 4. Incorporation of DCs results in increased propagation of multiple HIV-1 variants across IEDC co-cultures.** HIV-1 transmission by IE cultures and IEDC co-cultures, determined by luciferase activity of TZM-bl reporter cell line. IE cultures (open circles) or cognate IEDC co-cultures (closed circles) were apically exposed to CCR5-tropic (R5) HIV-1 NL4.3BaL, CXCR4-tropic (X4) NL4.3, transmitter/founder (T/F) virus CHO58 or T/F virus THRO for 72h. Subsequently, basolateral supernatants were collected and co-cultured for 48h with R5/X4 HIV-1 permissive TZM-bl luciferase reporter cell line. Data were normalized to non-infected (NI) sample. $n$ = 4–5 donors, Ratio paired $t$-test, $^*p<0.05$.

## Retention of replication-competent HIV-1 within intestinal epithelium is accompanied by preservation of barrier function

Targeting the early phases of mucosal HIV-1 infection within local tissue microenvironments represents a complex but essential step towards advancing HIV-1 curative therapies. Disruption of the epithelial barrier and establishment of cellular reservoirs across the mucosal barrier have been implicated in intestinal HIV-1 replication and long-term persistence of HIV-1 in infected tissues [6,43]. Here, we set out to investigate whether paracellular or transcellular HIV-1 transport through the intact epithelial barrier contribute to viral propagation upon intestinal HIV-1 invasion. We observed that junctional complex expression patterns at cell-cell borders, including tight junctions (zonula occludens-1, ZO-1), and adherence junctions (e-cadherin, E-cad) [Fig 5A], and FD4 permeation rate [Fig 5B] remained unaltered upon HIV-1 inoculation, in contrast to the positive control of TNF-α-induced disruption of intestinal epithelial integrity [S4A and S4B Fig]. Together, these findings indicate that apical exposure of intact intestinal epithelium to HIV-1 does not readily compromise tight junction function nor epithelial barrier permeability. The preservation of barrier function upon HIV-1 exposure contrasts with the disruptive impact of other enteric viruses to intact epithelium, such as SARS-CoV-2 [27], suggesting that concomitant microbial factors and inflammatory cascades likely contribute to the epithelial cell dysfunction observed in people living with HIV-1 [44,45]. Next, we assessed the ability of intestinal epithelium to retain infectious HIV-1 using a replication-competent, fluorescently-tagged HIV-1 [46]. Flow cytometry analyses showed capture of HIV-1 at 24h and 48h post-exposure [Fig 5C and 5D]. Notably, imaging of fluorescent viruses revealed an intracellular sequestration of replication-competent HIV-1 as a cluster of virions within epithelial cells for up to 2 days post-exposure [Fig 5E]. These data support a role for intact intestinal epithelium as cellular sanctuaries established early during acute infection enriched for replication-competent HIV-1 within human intestinal mucosa.

## Intraepithelial HIV-1 is targeted to LBPA+ multivesicular late endosomes

Upon the uptake of luminal components by epithelial cells, internalized material is trafficked intracellularly through endosomal pathways. This process involves the sorting of internalized content into distinct vesicular compartments, including early endosomes, recycling endosomes, or autophagosomes, and further processing in late endosomes/multivesicular bodies (MVBs) [47,48]. Previous research has demonstrated that HIV-1 can exploit endosomal trafficking pathways in both target cell lines and immune cells for viral replication [49–51]. Here, we aimed to define the intracellular trafficking pathway exploited by HIV-1 to penetrate the intestinal epithelium. To this end, we employed immunofluorescence imaging to monitor differential trafficking routes involving early endosomes, autophagy vesicles, recycling endosomes and late endosome/multivesicular bodies (MVBs) using the respective canonical vesicular markers: early endosome antigen 1 (EEA1), microtubule-associated 1A/1B-light chain 3 (LC3), ras-related protein Rab-11a (Rab11a) and lysobisphosphatidic acid (LBPA). At steady state, we observed minor expression levels of EEA1, LC3 and LBPA-positive vesicles alongside a great abundance of Rab11a+ vesicles [Fig 6A]. Upon HIV-1 inoculation, we observed an altered distribution of endocytic vesicular trafficking pathways within epithelial cells. In fact, a relatively minor increase of early endosomes and autophagy vesicles accompanied by a dramatic decrease in recycling endosomes and substantial enrichment of LBPA + MVBs was observed 24h post-HIV-1 exposure [Fig 6A]. Notably, co-staining for viral particles in combination with fluorescence intensity and Pearson's correlation coefficient analyses demonstrated colocalization of internalized HIV-1 particularly with LBPA+ MVBs at 24h post-exposure [Fig 6B–6D]. Furthermore, this LBPA+ intravesicular viral retention remained

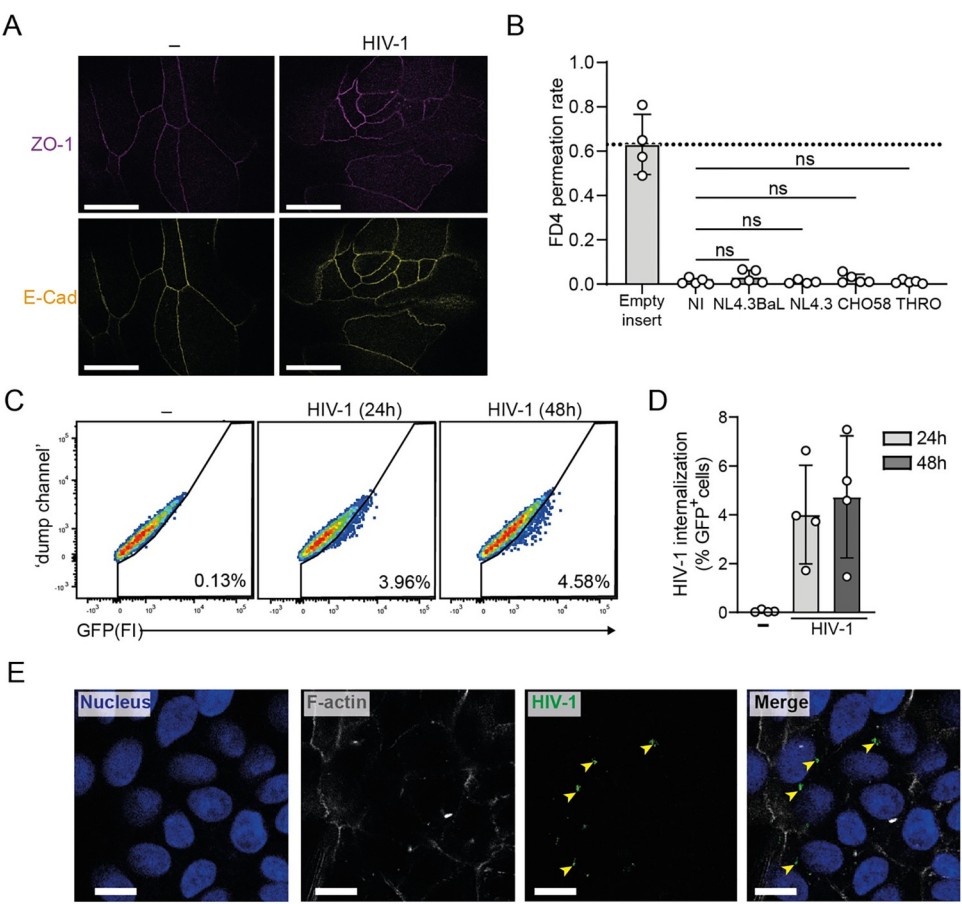

**Fig 5. HIV-1 is internalized by intestinal epithelial cells without disruption of intestinal barrier function.** (**A**) Junctional complexes in either untreated IE cultures or upon 72h exposure to HIV-1 NL4.3BaL by confocal microscopy analyses. Tight junction protein zonula occludens-1 (ZO-1, violet) and adherens junction adhesion protein e-cadherin (E-Cad, yellow). Scale bar = 20 micron, representative of $n$ = 2 donors. (**B**) Paracellular permeability of either non-infected (NI) intestinal epithelial monolayers or upon 72h exposure to multiple HIV-1 variants (R5) HIV-1 NL4.3BaL, CXCR4-tropic (X4) NL4.3, transmitter/founder (T/F) virus CHO58 or T/F virus THRO determined by 4 kDa FITC-conjugated dextran (FD4) permeation rate of empty insert or IE cultures at 4h post-FD4 addition. Permeability is expressed as FD4 permeation rate: FD4 basolateral$_{t = 4}$(μg)/FD4 apical$_{t = 0}$(μg). Dashed line (0.62) indicates the average of FD4 permeation rate from empty inserts. Data are mean ± SD of $n$ = 4–5 donors, One-way ANOVA, ns = non-significant. (**C,D**) HIV-1 uptake by intestinal epithelial cells, determined by monitoring % GFP fluorescence by flow cytometry analyses. (**C**) Representative flow cytometry plots and (**D**) quantification of epithelial monolayers exposed for 24h or 48h to replication-competent, fluorescently-tagged HIV-1 (HIV-1 Gag-iGFP). Data are mean ± SD of $n$ = 3 donors. (**E**) Confocal imaging of internalized HIV-1 Gag-iGFP (yellow arrowheads) within intestinal epithelial cells 48h post-exposure. Representative of $n$ = 2 donors. F-actin (grey), HIV-1 p24 capsid (green) and nuclei (DAPI, blue). Scale bar = 10 micron.

for up to 48h [S5 Fig]. These findings strongly suggest that HIV-1 follows a specialized LBPA-dependent transcellular trafficking route within human intestinal epithelium, and point towards subcellular LBPA+ vesicles as niches for infectious HIV-1 within intestinal mucosa.

## LBPA-dependent endosomal trafficking serves as an entry route for HIV-1 to replicate within intestinal mucosa *in vitro*

Having established that HIV-1 is targeted to LBPA+ vesicles within epithelial cells (Fig 6B–6D), and considering the reported LBPA functions in intracellular vesicular trafficking and as a

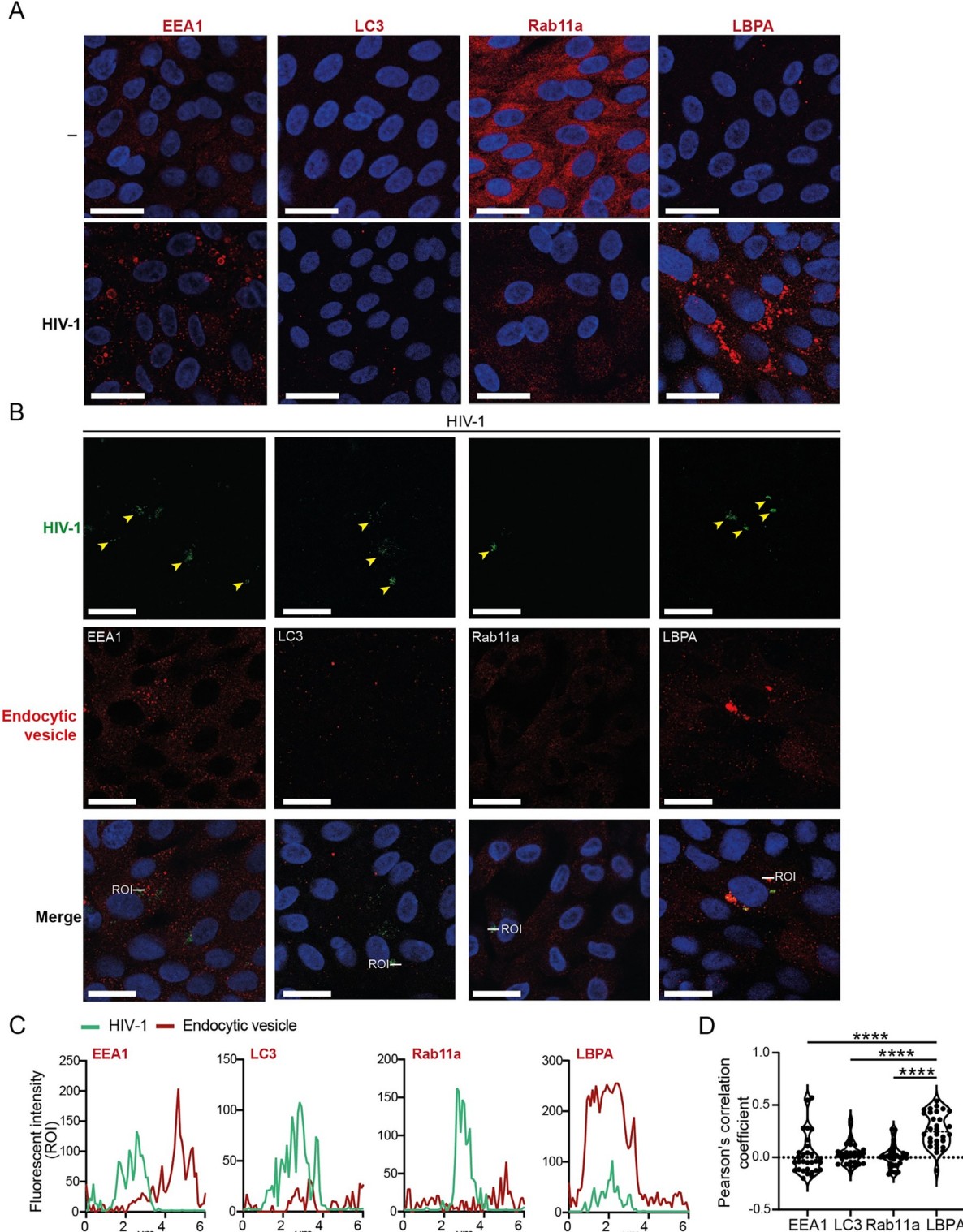

**Fig 6. Intestinal epithelial cells harbor HIV-1 within enriched multivesicular late endosomes.** (**A**) Distribution of endocytic vesicles in either untreated IE cultures or upon 24h exposure to HIV-1 Gag-iGFP by confocal microscopy analyses. Endocytic vesicle markers (red); Early endosomes (EEA1, Early endosome antigen 1), Autophagosomes (LC3, Microtubule-associated protein 1A/1B-light chain 3), Recycling endosomes (Rab11a, Ras-related protein Rab11a), Multivesicular late endosomes (LBPA, lysobisphosphatidic acid), and nuclei (blue). Scale bar = 20 micron, representative of $n$ = 3 donors. (**B**) Colocalization of endocytic vesicle markers in IE cultures upon 24h exposure to HIV-1

Gag-iGFP by confocal microscopy analyses. Endocytic vesicle markers (EEA1, LC3, Rab11a, LBPA, red), HIV-1 (green, filled arrowheads). Scale bar = 20 micron, representative of *n* = 3 donors. (**C**) Histograms of EEA1, LC3, Rab11a or LBPA and HIV-1 fluorescence intensities depicting the degree of overlap between HIV-1 and respective endocytic vesicle along the indicated 6 micron-long Region of Interest (ROI, as specified in B). (**D**) Pearson's correlation coefficient for colocalization analyses between each endocytic vesicle marker and HIV-1, *n* = 3 donors, Two-way ANOVA, **** *p* <0.0001.

modulator of the formation of MVBs [52], we next sought to assess the role of LBPA-dependent trafficking in establishment of intestinal HIV-1 infection. To accomplish this, we pre-treated IE cultures or IEDC co-cultures for 16h with monoclonal anti-LBPA, which has previously been shown to neutralize LBPA functionality [17–19]. We confirmed reduction of multivesicular late endosomes in intestinal epithelial cells upon anti-LBPA treatment (Fig 7A and 7B). Markedly, deactivation of LBPA limited HIV-1 transmission mediated by IE cultures as well as by IEDC co-cultures[Fig 7C–7F], which supports a role for intraepithelial LBPA-mediated endosomal pathways in driving enteric HIV-1 replication and transmission. Our study thereby identifies the LBPA-directed MVB trafficking route as a transcellular release mechanism of infectious HIV-1 within human intestinal mucosa. Furthermore, these data substantiate the relevancy of human intestinal infection organoid models in unveiling mechanisms of HIV-host interactions within tissue microenvironments *in vitro*, and underscore the therapeutic potential of LBPA-targeting strategies to suppress local intestinal HIV-1 infection.

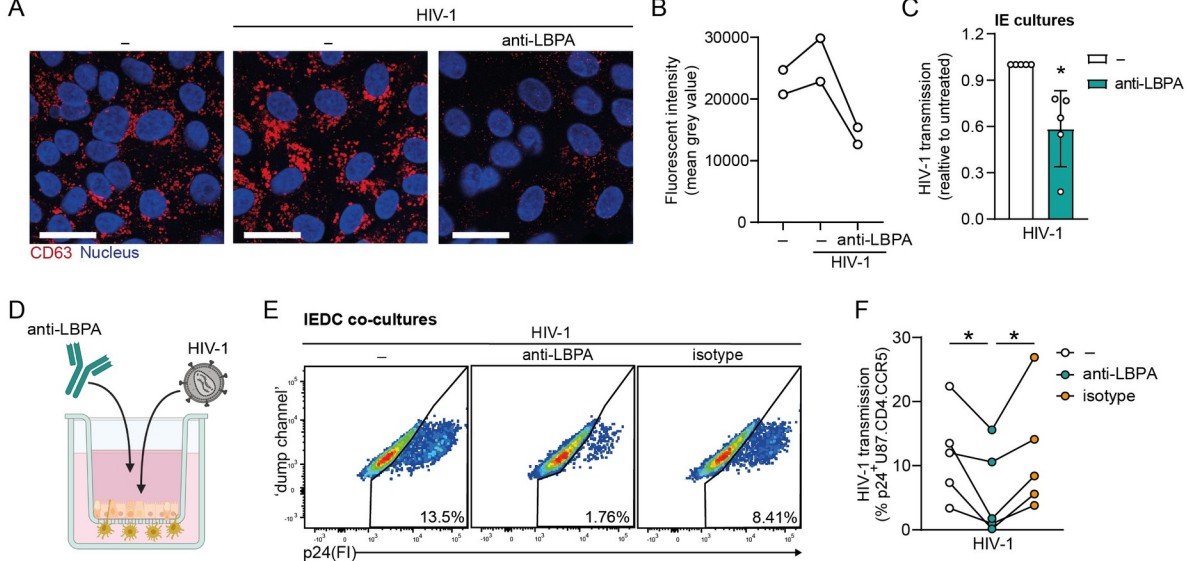

**Fig 7. Neutralization of LBPA-mediated endocytic pathway curbs intestinal HIV-1 transmission *in vitro*.** (**A,B**) Distribution of multivesicular/late endosomes in IE cultures pre-treated with either neutralizing anti-LBPA antibody (15 µg/ml), or left untreated for 16 h, followed by 24h exposure to HIV-1 Gag-iGFP visualised by confocal microscopy analyses. (**A**) Multivesicular/late endosome marker (CD63, red) and nuclei (blue). Scale bar = 20 micron, representative of *n* = 2 donors. (**B**) Analyses of fluorescence intensity were performed at original magnification by measuring mean grey value with ImageJ software [22]. Open circles represent averages derived from 5 different fields of view, *n* = 2 donors. (**C-F**) HIV-1 transmission by IE cultures or IEDC co-cultures prophylactically-treated with neutralizing anti-LBPA antibody. IE cultures and IEDC co-cultures were apically pre-treated with either 15 µg/ml anti-LBPA antibody, 15 µg/ml isotype antibody control or left untreated for 16 h, followed by exposure to HIV-1 NL4.3BaL for 72h. Subsequently, basolateral supernatants were collected and co-cultured with HIV-1 permissive U87.CD4.CCR5 target cells for 96h. HIV-1 transmission was determined by intracellular p24 staining of U87.CD4.CCR5 cell line by flow cytometry analyses. (**C**) HIV-1 transmission by IE cultures. Data were normalized to untreated samples and are mean ± SD of *n* = 5 donors, One sample *t*-test, **p* = 0.0196. (**D**) Schematic representation of IEDC co-culture model as drug screening platform for intestinal HIV-1 infection *in vitro*. (**E**) Representative flow cytometry plots and (**F**) quantification of HIV-transmission by IEDC co-cultures. *n* = 5 donors, Two-way ANOVA, **p*<0.05.

## Discussion

Although the gastrointestinal tract is a key player in HIV-1 acquisition and HIV-1 disease pathogenesis, existing *in vitro* models for studying the mechanisms underlying mucosal viral transmission are both limited and suboptimal in recapitulating key factors for intestinal HIV-1 infection in humans. In this study, we present a human 2D primary immune-competent organoid model consisting of intestinal epithelium co-cultured with DCs, and have demonstrated that it represents a suitable *in vitro* system to study epithelium-immune interactions in intestinal HIV-1 transmission. Importantly, immune cells are the primary cellular targets for HIV-1 infection and dissemination [53–57], however recent literature has shown that other cell types, such as microglia [58,59] and various tissue-derived epithelial cells, are also targeted by HIV-1. For example, HIV-1 can infect and establish latency in *in vitro* lung bronchial epithelial cells, which can be reactivated, leading to resumption of HIV-1 replication upon inflammation [60,61]. Additionally, HIV-1 penetrates renal, tonsil, foreskin, and cervical epithelial cells to subsequently spread to activated immune target cells via cell-to-cell contact [55,62]. Herein we demonstrated that human intestinal epithelial cells internalize and transmit infectious HIV-1 virions to permissive cells not only via direct cell-cell contact, but also in a cell-free manner without disruption of epithelial barrier integrity. Evolvement of the IEDC model with inclusion of additional relevant compartments such as microbiome, mucus layer and inflammatory cues will enable the study of HIV-1 interaction with epithelium at a more complex *in vitro* level.

Furthermore, the retention of internalized HIV-1 virions within epithelial cells in combination with the unscathed intestinal barrier integrity post-HIV-1 exposure pointed towards transcellular transmission of the virus rather than extracellular transport through paracellular spaces. We therefore sought to map the intra-endosomal trafficking of captured HIV-1. We observed that HIV-epithelium interaction impacted the distribution of endocytic vesicular pathways accompanied with targeting of intraepithelial HIV-1 particularly to LBPA+ multivesicular late endosomes [55,63]. Follow-up studies are required to determine whether LBPA-rich vesicle reorganization is directly mediated by viral components or elicited during antiviral immune responses by epithelial cells. Notably, suppression of viral transmission by our IE culture and IEDC coculture models after antibody-mediated inactivation of LBPA indicated the virus-facilitating role of the subcellular LBPA-dependent trafficking mechanism in intestinal HIV-1 dissemination. LBPA-mediated transcellular transport across the intestinal epithelial barrier may have additional implications beyond those showcased in the current study, including initial establishment of gut-associated viral reservoirs within the subepithelial mucosal myeloid and T cell compartments and HIV-1 associated immune activation. Future studies using our innovative IEDC model could also be employed to investigate the functional impact of extrinsic factors such as microbiome dysbiosis or mucosal immune activation on LBPA-mediated HIV-1 transcellular transport and residual replication, as well as on HIV-1 persistence and viral rebound within intestinal tissue microenvironment *in vitro*.

Furthermore, herein, we have demonstrated that HIV-1 targets subepithelial DCs in intestinal immune-competent infection organoid model for mucosal viral dissemination. Upon apical exposure of the IEDC co-culture model to HIV-1, protrusions from basolateral DCs were able to capture HIV-1 across the epithelial barrier resulting in amplification of local viral replication. The enhanced HIV-1 transmission rate by the IEDC co-culture model compared to that of cognate IE cultures highlights both the key role of resident immune cells in establishment of intestinal HIV-1 infection as well as the pertinence of using human models encompassing multiple cell types for studying viral pathogenesis. Due to the paucity of intestinal-derived DCs, the current study utilized the readily accessible allogeneic blood monocyte-

derived DCs for the IEDC co-culture model, which allows more robust experimental reproducibility of the model. However, we cannot exclude that allogenic responses and HLA-mismatch could impact epithelial-DC immune interactions [64,65]. In future research, we will test the co-culture abilities of lamina propria-derived cellular targets in the IE culture model to reproduce autologous epithelial-immune interactions, thus further increasing the verisimilitude of *in vitro* intestinal mucosal infection organoid models. Additionally, use of such models that more closely resemble *in vivo* tissue multicellular complexity allows for experimental manipulation and thereby hold great potential for pre-clinical drug screening, as illustrated by our findings that targeting host molecule LBPA suppresses HIV-1 transmission across IEDC co-cultures. We envision that the model we present herein can be utilized for studying tissue-resident immune responses in the context of not only gastrointestinal infections but also auto-immune/inflammatory diseases. Our findings add to the growing body of literature demonstrating that combinatorial strategies with host-directed therapeutics such as targeting intraepithelial LBPA, particularly during the early phases of acute HIV-1 infection to impact HIV-1 persistence and replication within mucosal myeloid compartment in secluded tissue compartments, could represent an advancement in HIV-1 curative therapies [12,27]. In summary, our findings identify LBPA+ vesicle-mediated transcellular route in the human intestine as a novel mechanism of mucosal HIV-1 entry and subepithelial DC-mediated viral dissemination. Furthermore, we present an *in vitro* reconstructed intestinal mucosal system for modelling viral diseases that will facilitate future investigation into the cellular and molecular mechanisms of not only HIV-1 disease, but also other intestinal chronic viral infections such as SARS-CoV-2, as well as serve as a platform for antiviral drug discovery.

## Materials and methods

### Ethics statement

Fetal material was used as determined by Dutch law (Wet Foetal Weefsel), which states that human fetal tissues/cells can only be used for medical purposes, or medical and scientific research. Human fetal intestinal tissue samples (gestational age 13–18 weeks) were obtained from the HIS (Human Immune System) Mouse Facility of the Amsterdam UMC (University medical Center), Amsterdam. All material has been collected from donors from whom written informed consent has been obtained for the use of the material for research purposes. The fetal donor information is anonymized and is not available to the Amsterdam UMC. Buffy coats derived from blood donations (Sanquin blood bank, the Netherlands) were obtained with approval of the Medical Ethics Review Committee of the Amsterdam UMC. Buffy coats were handled in accordance with the relevant guidelines and regulations, as stated in the Amsterdam UMC Research Code. Use of buffy coats is not subjected to informed consent according to the Medical Research Involving Human Subjects Act and the Medical Ethics Review Committee of Amsterdam UMC.

### Isolation of intestinal epithelium-derived cells and lamina propria-derived cells

Intestinal epithelium-derived cells and lamina propria-derived cells were obtained as previously described [12,66]. Human fetal intestinal were cut open longitudinally and cut into 0.5x0.5 cm pieces and extensively washed with ice-cold PBS. Subsequently, the epithelial layer was detached by digesting the intestinal tissue pieces in PBS supplemented with 5 mM EDTA (Sigma-Aldrich), 2 mM DTT (Sigma-Aldrich) and 1% FCS (Bio-connect) for 2x 20 minutes at 4˚C. Epithelium-derived cells were isolated from the supernatant by straining it through a

70 μm cell strainer (Falcon, Corning) followed by washing steps to obtain a single cell solution. Single cells were cultured in differentiation medium (DM), consisting of a 1:1 mixture of IntestiCult organoid growth medium basal medium (OGM; STEMCELL Technologies) and Advanced DMEM/F12 (Thermo Fisher Scientific) supplemented with 10 mM GlutaMAX (Thermo Fisher Scientific), 10 mM HEPES (Sigma), and Penicillin/Streptomycin (10 U/mL and 10 μg/ml, respectively; Invitrogen), referred to as Advanced+++ medium from hereon. Alternatively cells were cultured in a 1:1 mixture of IntestiCult organoid differentiation medium (STEMCELL Technologies) and organoid supplement (STEMCELL Technologies) and Penicillin/Streptomycin. In order to harvest lamina propria cells, remaining intestinal tissue devoid of the epithelial layer was minced and digested for 2 × 30 minutes at 37˚C with Iscoves Modified Dulbecco's Medium (IMDM, Thermo Fischer Scientific, USA) supplemented with 1 mg/mL (0.15 U/mg) Collagenase D (Roche), penicillin/streptomycin (10 U/mL and 10 μg/ml, respectively; Invitrogen), 1% FCS, and 1000 U/mL DNAse type I (Worthington Biochemical Corporation). To obtain a single cell suspension, the resultant supernatant containing lamina propria cells was filtered through a 70 micron strainer. Lamina propria-derived mononuclear cells were isolated from the single cell suspension using Lymphoprep gradient (Axis-Shield), and cultured in IMDM supplemented with 10% FCS, penicillin/streptomycin (10 U/mL and 10 μg/ml, respectively), and 50 U/mL IL-2 (Miltenyi) for use in HIV-1 infection experiments.

## IE cultures

Human fetal intestinal organoids, were used to generate IE culture systems as described previously [26,27,66]. Briefly, following isolation of a single cell suspension of epithelium-derived cells as described above, cells were suspended in a 1:3 mixture of Advanced+++ medium: Matrigel (Corning) and seeded in a 24-well plate (Corning). Intestinal organoid cultures were maintained in expansion medium (EM) consisting of OGM basal medium and organoid supplement (STEMCELL Technologies) in a 1:1 ratio supplemented with penicillin/streptomycin (10 U/mL and 10 μg/ml). Medium was additionally supplemented with 10 $\mu$M Y-27632 (RHO/ROCK pathway inhibitor; STEMCELL Technologies) for the first 3 days after seeding. Medium was refreshed every 2–3 days and organoids were passaged once per week by mechanical disruption. Intestinal organoid cultures were kept at 37˚C and 5% $CO_2$ and subsequently used to generate human IE cultures as previously described [26,27]. 3.0 μm pore 24-well cell culture inserts (CellQart, SABEU) were coated with 20 μg/mL rat collagen type 1 (Ibidi) in 0.1% acetic acid (Sigma Aldrich). Single cells were obtained from intestinal organoid cultures using TrypLE (Gibco, Thermo Fischer Scientific) digestion. $1*10^5$ single cells were suspended in EM supplemented with Y-27632 and added onto the collagen coated cell culture inserts. For the first 5–7 days the monolayers were cultured in EM (and supplemented with Y-27632 for 3 days). From day 5–7 onwards all monolayers were cultured in DM, and refreshed every 3–4 days.

## Generation and phenotypic characterization of monocyte-derived DCs

Monocyte-derived DCs were generated either by gradient centrifugation as previously described [12], or using positive magnetic-activated cell-sorting for CD14+ monocytes as per the manufacturers' instructions (Miltenyi). Briefly, PBMCs were isolated from buffy coats of healthy donors (Sanquin) using a Lymphoprep (Axis-Shield) gradient. Subsequently, monocytes were enriched using either a Percoll (Amersham Biosciences) gradient step or by positive immunomagnetic selection with CD14 Microbeads (Miltenyi, 130-050-201). Isolated monocytes were then differentiated into immature DCs over 4–6 days in RPMI 1640 containing

10% FCS, penicillin/streptomycin (10 U/mL and 10 μg/ml, respectively), 2 mM L-glutamine (Lonza), 500 U/mL IL-4 (Invitrogen), and 800 U/mL GM-CSF (Invitrogen). These experimental protocols consistently generate high-purity viable, immature monocyte-derived DCs expressing prototypical DC surface markers CD11c and DC-SIGN, and devoid of activation/maturation markers such CD83 and CD86, as determined by flow cytometry analyses [S2A–S2F Fig] [11,12]. Phenotypic characterization of isolated monocytes and subsequently differentiation to monocyte-derived DCs was determined by staining cells with CD45-BV711 (clone HI30, Biolegend), CD14-PE-Dazzle594 (clone HCD14, Biolegend), CD11c-PE-Cy7 (clone B-ly6, BD Pharmigen), and DC-SIGN (clone 120612, Biotechne). Stained cells were acquired with FACSymphony A1 flow cytometer (BD Biosciences) and analyzed with FlowJo version 10.6.2. DC activation/maturation at steady-state and upon treatment for 48h LPS (10 ng/mL, Lipopolysaccharides from Salmonella Typhimurium, Sigma-Aldrich) was determined by staining cells with CD83-AF647 (clone HB15e, Biolegend) and CD86-AF488 (clone IT2.2, Biolegend) in PBS/BSA buffer. Stained cells were acquired with FACS Canto II flow cytometer (BD Biosciences) and analyzed with FlowJo version 10.6.2.

## Cell viability

Cell viability was measured using the ATP-based CellTiter-Glo 3D reagent (Promega) according to the manufacturer's instructions. As previously described [27], the CellTiter-Glo 3D reagent and white 96-well plate were equilibrated to RT. Medium was removed from the DCs and then 100 ul of 1:1 ratio of CellTiter-Glo 3D reagent and dPBS (Lonza) was added to the cells and mixed vigorously. After incubating for 15 minutes at 37˚C to induce cell lysis, the suspension was transferred to a white 96-well plate and luminescence was measured by the SynergyHTX (BioTek) plate reader and expressed in relative light units (RLU).

## IEDC co-cultures

Polarized IE cultures were grown on collagen type l-coated 3.0 micron pore cell culture inserts for 10–13 days as aforementioned. IE cultures deemed polarized (by imaging microscopy analyses), confluent (TEER value above 200 $\Omega.cm^2$) and impermeable (by FITC-conjugated dextran permeation rates from apical to basolateral compartments) were used to establish the human intestinal epithelium-DC (IEDC) co-cultures. Subsequently, the polarized, tight and non-permeable IE cultures on cell culture inserts were inverted and placed in a 6-well plate and $2*10^5$ immature monocyte-derived DCs suspended in 50 μl DM supplemented with 800 U/mL GM-CSF was seeded onto the underside (basolateral) of the inverted inserts. Attachment of DCs took place at 37˚C with 5% $CO_2$ for 2h. After attachment, inserts with co-cultures of basolateral-attached DCs and apical differentiated epithelial monolayers, were inverted back into a 24 well plate and cultured in DM supplemented with 800 U/mL GM-CSF at both the apical and basolateral compartments. IEDC co-cultures were allowed to equilibrate at 37˚C 5% $CO_2$ for 24h, and the basolateral compartment was subsequently washed to remove any unattached DCs prior to use in HIV-1 infection experiments.

## Transepithelial electrical resistance measurement

The integrity of the monolayer intestinal barrier was determined by transepithelial electrical resistance (TEER) measurement using an Epithelial Volt/Ohm meter (EVOM2). TEER values $(Ohm.cm^2)$ were calculated as described in [67]. Monolayers were deemed sufficiently formed when the TEER was over 200 $Ohm.cm^2$. To induce experimental disruption of barrier integrity, IE cultures grown in 0.4 μm pore 24-well cell culture inserts (Corning) were treated at the

basolateral compartment with recombinant human TNF-α protein (400 ng/mL, Bio-Techne) for 96h.

## FITC-dextran paracellular permeability

The permeability of the intestinal epithelial monolayer was assessed using FITC-conjugated dextran (4kDa FD4, Sigma Aldrich) translocation from the apical to the basolateral compartment as previously described [26,27]. IE cultures were washed with Hanks' Balanced Salt Solution without phenol-red (HBSS, Lonza) and incubated apically with 1mg/mL FITC-dextran solution for 4h. The amount of FITC-dextran in both apical and basolateral chambers was determined using a BioTek Synergy HT plate reader. FD4 concentrations were determined using standard curves. Permeability is expressed as FD4 permeation rate: FD4 basolateral$_{t\,=\,4}$(μg)/FD4 apical$_{t\,=\,0}$(μg).

## U87.CD4.CCR5 cell line

U87 cell lines stably expressing CD4 and wild-type CCR5 co-receptor was obtained through the NIH AIDS Reagent Program, Division of AIDS, NIAID, NIH: U87.CD4.CCR5. cells (Cat# ARP-4035) from H. K. Deng and D. R. Littman [68] and maintained in IMDM supplemented with 10% FCS and penicillin/streptomycin (10 U/mL and 10 μg/ml, respectively).

## HIV-1

HIV-1 NL4.3BaL, NL4.3, CHO58, THRO, and HIV Gag-iGFP were produced as previously described [11,69,70]. T/F plasmids for CHO58 and THRO were obtained through the NIH HIV Reagent Program, NIAID, NIH: Panel of Full-Length Transmitted/Founder (T/F) Human Immunodeficiency Virus Type 1 (HIV-1) Infectious Molecular Clones, HRP-11919, contributed by Dr. John C. Kappes. HIV Gag-iGFP plasmid was obtained through the NIH HIV Reagent Program, Division of AIDS, NIAID, NIH: Human Immunodeficiency Virus (HIV) Gag-iGFP, ARP-12457, contributed by Dr. Benjamin Chen. Plasmids were amplified by transformation into STBL3 *E.coli* bacteria (Invitrogen), and 293T cells were subsequently transfected with proviral plasmids. Viruses were harvested at day 2 and 3. TCID50 was determined using TZM-bl indicator cells (John C. Kappes, Xiaoyun Wu, Birmingham, Alabama, USA and Tranzyme Inc., the NIH AIDS Reagent Program, Division of AIDS, NIAID) [11].

## HIV-1 transmission by human intestinal-derived cells

Human fetal intestinal epithelium-derived cells and lamina propria mononuclear cells were isolated as described above. Subsequently, co-isolated intraepithelial HIV cellular targets were depleted by sorting for epithelium-derived CD45⁻ cells using 4-laser FACS SONY SH800 (nucleated hematopoietic cell marker CD45-FITC, (clone HI30, eBioscience)). Intestinal epithelium-derived CD45⁻ cells or lamina propria-derived cells supplemented with IL-2 (50U/ml; Miltenyi) were pre-treated with 50, 25, 12.5 or 6.25 μM tenofovir (TFV, ARP-10199) 30 μM Maraviroc (MCV, ARP-11580), 1 μM Emtricitabine (FTC, HRP-10071), 1 μM Indinavir (IDV, ARP-8145), or a combinatory pre-treatment of 25 μM TFV with 0.2 μM FTC or 0.2 μM IDV for 2h, prior to HIV-1 NL4.3BaL inoculation (MOI = 0.13). All antiviral reagents were obtained through the NIH HIV Reagent Program, Division of AIDS, NIAID, contributed by DAIDS/NIAID. 48h after HIV-1 inoculation, cells were extensively washed to remove unbound virus, and co-cultured with the HIV-1 permissive U87.CD4.CCR5 cell line for an additional 72h. HIV-1 transmission by epithelium-derived and lamina propria-derived cells was determined by intracellular p24 staining by flow cytometer. DC marker CD11c (clone B-

ly6-PE-Cy7, BD Pharmigen) staining and T cell CD3 (clone UCTH1-APC-FIRE750, Biole-gend) were used to exclude intestinal HIV cellular targets from analysis in the co-culture. HIV-1 infection of U87.CD4.CCR5 in co-culture with intestinal-derived cells was thereby defined as CD11c⁻CD3⁻p24⁺ cells and is representative of HIV-1 transmission by intestinal-derived cells.

## HIV-1 transmission by intestinal organotypic cultures

IE cultures and IEDC cocultures were inoculated apically with HIV-1 strains NL4.3BaL (100 μl/insert, TCID50 $187*10^3$, determined by TZM-bl cells), NL4.3 (100 μl/insert, TCID50 $139.8*10^3$, determined by TZM-bl cells), CHO58 (100 μl/insert, TCID50 $25*10^3$, determined by TZM-bl cells), or THRO (100 μl/insert, TCID50 $7.48*10^3$, determined by TZM-bl cells) and subsequently incubated for 3 days at 37˚C with 5% CO2. When indicated, cultures were pre-treated apically with either neutralizing anti-LBPA antibody (15 μg/mL, clone 6C4, Sigma-Aldrich), IgG1 isotype antibody control (15 μg/ml, clone P3.6.2.8.1, Invitrogen) for 16h. Baso-lateral supernatants were harvested 72h post viral exposure and added onto the CCR5 and CXCR4 HIV-1-permissive TZM-bl reporter cell line for 2 days at 37˚C with 5% CO2 or CCR5 HIV-1 permissive U87.CD4.CCR5 cell line for 4 days), at 37˚C with 10% CO2. HIV-1 infec-tion of TZM-bl cell luciferase was measured using britelite plus Reporter Gene Assay System (Revvity). HIV-1 infection of U87.CD4.CCR5 cells was determined via intracellular staining of the HIV-1 capsid protein p24 by flow cytometry. Briefly, the U87.CD4.CCR5 cells were fixed with 4% paraformaldehyde (PFA; Electron Microscopy Sciences) for 30 minutes at room tem-perature, washed with PBS and then permeabilized with PBS supplemented with 0.5% saponin (Sigma) and 1% bovine serum albumin (BSA; Roche). Next, the cells were stained with Mouse IgG1 anti-p24 (clone KC57-PE, Beckman) in PBS supplemented with 0.5% saponin and 1% bovine serum albumin (BSA; Roche) for 30 minutes at 4˚C. Stained cells were acquired with FACS Canto II flow cytometer (BD Biosciences) and analyzed with FlowJo version 10.6.2. HIV-1 infection of TZM-bl reporter cell line or U87.CD4.CCR5 cells is thereby representative of basolateral infectious virus release by intestinal organotypic cultures.

## HIV-1 internalization

IE cultures cultured unto collagen type l-coated 96-well plates (Corning) or polymer μ-slide 8 well (Ibidi) were exposed to replication-competent HIV Gag-iGFP (100 μl/well, TCID50 $5*10^3$, determined by TZM-bl cells), for 24h or 48h. For assessment of HIV-1 internalization by flow cytometry analysis, monolayers were digested into single cells using Advanced+++ medium supplemented with 1 mg/mL (0.15 U/mg) Collagenase D, and penicillin/streptomy-cin (10 U/mL and 10 μg/ml, respectively) at 37˚C for 15 min, and thereafter fixed in 4% PFA for 30 minutes at RT. For assessment of HIV-1 internalization by confocal microscopy analy-sis, IE cultures were washed to remove unbound virus and thereafter fixed in 4% PFA for 30 minutes at RT. Quantification of HIV-1 internalization was assessed by monitoring GFP fluo-rescence in fixed cells using FACS Canto II flow cytometer (BD Biosciences) and analyzed with FlowJo version 10.6.2, or visualized using confocal imaging microscopy using a Leica TCS SP8 X mounted on a Leica DMI6000.

## HIV-1 infection and activation of basolateral detached DCs

Basolateral DCs were harvested from HIV-1-NL4.3BaL infected IEDCs 72h post-infection (pi) by incubating the basolateral compartment with Accutase Cell Detachment Solution (STEM-CELL Technologies) for 10 minutes at 37˚C. Subsequently, suspension was aspirated and dis-pensed multiple times to detach basolateral DCs. The cells were fixed with 4% PFA for 30

minutes at RT, washed with PBS and then permeabilized with PBS supplemented with 0.5% saponin and 1% BSA. Next, the cells were stained with CD45-BV711 (clone HI30, Biolegend), CD11c-PE-Cy7 (clone B-ly6, BD Pharmigen), DC-SIGN-AF750 (clone 120612, Biotechne), and Mouse IgG1 anti-p24 (clone KC57-PE, Beckman) for 30 minutes at 4˚C in PBS supplemented with 0.5% saponin and 1% BSA. Stained cells were acquired with FACS Canto II flow cytometer (BD Biosciences) and analyzed with FlowJo version 10.6.2. HIV-1 infection of detached basolateral DCs was defined as CD11c$^+$DC-SIGN$^+$p24$^+$ cells. HIV-induced DC activation was determined in IEDCs 72h post HIV-1 exposure by surface staining of basolateral detached DCs with CD83-AF647 (clone HB15e, Biolegend) and CD86-AF488 (clone IT2.2, Biolegend) for 30 minutes at 4˚C in PBS and cells were acquired with FACSymphony A1 flow cytometer (BD Biosciences) and analyzed with FlowJo version 10.6.2.

## HIV-1 transmission by basolateral detached basolateral DCs

Basolateral DCs were harvested from HIV-1-NL4.3BaL infected IEDCs 72h pi as described above. DCs were extensively washed to remove input virus, and co-cultured with the HIV-1 permissive U87.CD4.CCR5 cell line for an additional 72h. HIV-1 transmission by DCs was determined by intracellular p24 staining by flow cytometer. DC marker CD11c-PE-Cy7 (clone B-ly6, BD Pharmigen) staining was used to exclude DCs from analysis in the co-culture [12]. HIV-1 infection of U87.CD4.CCR5 in co-culture with DCs was thereby defined as CD11c-p24 + cells. Stained cells were acquired with FACS Canto II flow cytometer (BD Biosciences) and analyzed with FlowJo version 10.6.2.

## Confocal imaging microscopy

Intestinal epithelial monolayers were fixed in 4% PFA for at least 30 minutes at RT, and then washed three times with dPBS (Lonza Bioscience) preceding storage at 4˚C until staining. Membranes were cut out of the insert and permeabilized with 0.5% Triton-X100 (Sigma-Aldrich) for 10 minutes, washed three times with dPBS, and subsequently blocked with fish serum blocking buffer (Thermofisher) for 1h at RT. Cells were incubated overnight at 4˚C with primary antibodies diluted in fish serum blocking buffer. Primary antibodies included mouse IgG1 anti-p24 (clone KC57, Beckman Coulter), mouse IgG2b anti-CD11c (clone S-HCL-3, BD bioscience), mouse IgG1 anti-LC3 (clone 4E12, MBL), rat IgG2a anti-ZO-1 (clone R40.76, Santa Cruz biotechnology), mouse IgG2b anti-E-Cad (clone 36/E-Cadherin (RUO, BD Bioscience), mouse IgG1 anti-LBPA (clone 6C4, Sigma Aldrich), polyclonal rabbit anti-Rab11a (71–5300, Invitrogen), mouse IgG1 anti-EEA1 (clone 14/EEA1 (RUO), BD bioscience), mouse IgG1 anti-CD63 (clone MEM-259, Invitrogen) and mouse IgG2a anti-GFP (clone B2, Santa Cruz Biotechnology). Stained cells were washed three times and incubated with secondary antibodies diluted at 1:400 together with the probe Phalloidin CruzFluor-488, -555, or -647 (respectively sc-363791, sc-363794, sc-363797, Santa Cruz Biotechnology) diluted at 1:1500 in fish serum blocking buffer for 1h at RT. Secondary antibodies included goat anti-mouse IgG1 Alexa 647 (A-21240, Invitrogen), goat anti-rabbit IgG Alexa 594 (A-11012, Invitrogen), goat anti-mouse IgG2b Alexa 546 (A-21143, Invitrogen), goat anti-rat IgG 488 (A-11006, Invitrogen), and goat anti-mouse IgG2a Alexa 488 (A-21131, Invitrogen). Nuclei were stained with 300nM DAPI (Invitrogen) for 10 minutes at RT. Cells were subsequently washed three times in dPBS and mounted in Prolong Gold Diamond Antifade mountant (Invitrogen). Samples were imaged on a Leica TCS SP8 X mounted on a Leica DMI6000 and analyzed using LAS X (Leica application Suite X, Leica microsystems). Fluorescence intensities were measured across a set distance within a ROI, 10 ROIs per donor, using LAS X. Quantitative analysis of fluorescence intensity was performed by measuring the mean grey value with ImageJ software [27].

## Statistical analyses

Statistical analyses were performed using GraphPad Prism 9 (Graphpad Software, Inc.). For paired observations, two-tailed parametric Student's $t$-tests were performed. For paired observations with non-consistent measures of effect, ratio parametric Student's $t$-tests were performed. For experiments in which data consists of multiple comparisons with a single independent variable, a one-way analyses of variance (ANOVA) were performed. For experiments in which data consists of multiple groups, two-way analyses of variance (ANOVA) were performed. For data shown in relative, data were normalized to untreated-virus infected samples (set at 1). A one-sample $t$-test was then utilized to compare fold changes in experimental conditions (treated-virus infected samples) to the hypothetical population mean of 1 [27]. Pearson's correlation coefficient was calculated using LAS X. The threshold for statistical significance was set at $^*P < 0.05$, $^{**}P < 0.01$, $^{***}P < 0.001$.

## Supporting information

**S1 Fig. Suppression of HIV-1 replication in U87.CD4.CCR5 cell line by antiretroviral therapeutics.** Productive HIV-1 infection of U87.CD4.CCR5 cell line, determined by intracellular p24 staining by flow cytometry analyses. U87.CD4.CCR5 cells were pre-treated with nucleotide reverse-transcriptase inhibitor Tenofovir (TFV, 50 μM), CCR5-antagonist Maraviroc (MCV, 30 μM), nucleoside reverse-transcriptase inhibitor Emtricitabine (FTC, 1 μM), protease inhibitor Indinavir (IDV,1 μM), or a combination of TFV (25 μM) with either FTC (0.2 μM) or IDV (0.2 μM) for 2h or left untreated prior to exposure to HIV-1 NL4.3BaL for 72h. Open circles represent $n$ = 2 biological replicates.
(TIF)

**S2 Fig. Phenotypic characterization, cell viability and productive HIV-1 infection of monocyte-derived DCs used to establish IEDC co-cultures.** (**A-C**) Purity of monocyte-derived DC differentiation was routinely determined by surface staining for nucleated hematopoietic cell marker CD45, monocyte marker CD14 and DC markers CD11c and DC-SIGN by flow cytometry analyses. Representative flow cytometry plots of the (**A**) phenotypic characterization of affinity purified CD45$^+$CD11c$^-$DC-SIGN$^-$CD14$^+$ human monocytes from PBMCs by positive immunomagnetic selection using CD14 Microbeads and (**B**) differentiation of the isolated monocytes to CD45$^+$CD11c$^+$DC-SIGN$^+$ DCs (on day 4 of differentiation as used to establish the IEDC co-cultures). (**C**) Quantification of purity of monocyte-derived CD11c$^+$DC-SIGN$^+$ DCs prior to establishment of IEDC co-cultures. Data are mean ± SD of $n$ = 3 donors, Student's paired $t$ test. $^{**}p<0.01$. (**D,E**) Maturation of monocyte-derived DCs was determined by surface staining for canonical activation DC markers CD83 and CD86 by flow cytometry analyses. (**D**) Representative flow cytometry plots and histograms of untreated DCs (as used to establish the IEDC co-cultures) or DCs treated with lipopolysaccharide (LPS,10 ng/mL) for 48h. LPS-treated DCs displayed a mature phenotype, in contrast to untreated DCs, as evident by increased expression of maturation markers CD83 and CD86. (**E**) quantification of flow cytometry data. $n$ = 2 donors. (**F**) Viability of DCs cultured in different media formulations, determined by ATP-Based CellTiter-Glo luminescent cell viability assay. DCs were either cultured 24h in standard RPMI-based DC culture medium (i.e. RPMI 1640 supplemented with 10% FCS, 10 U/ml and 10 μg/ml, penicillin/streptomycin respectively, 2 mM L-glutamine, 500 U/mL IL-4 and 800 U/mL GM-CSF) or standard organoid differentiation medium supplemented with GM-CSF. Data is mean of $n$ = 2 donors (RLU, relative light units). (**G**) Productive HIV-1 infection of DCs, determined by intracellular p24 staining by flow cytometry analyses. DCs were pre-treated with nucleotide reverse-

transcriptase inhibitor Tenofovir (TFV, 25 μM) or nucleoside reverse-transcriptase inhibitor Emtricitabine Emtricitabine (FTC, 1 μM) for 2h or left untreated prior to exposure to HIV-1 NL4.3BaL for 6 days. Data are mean ± SD of $n = 3$ donors, One-way ANOVA, *$p < 0.05$.
(TIF)

**S3 Fig. Basolateral detached DCs from IEDC co-cultures are infected with HIV-1 and subsequently mediate HIV-1 transmission via cell-cell contact to target U87.CD4.CCR5 cell line.** (A) Gating strategy to determine HIV-1 infection of detached basolateral CD11c⁺DC-SIGN⁺ DCs derived from non-infected versus HIV-1 exposed IEDC co-cultures using multiparameter intracellular p24 staining by flow cytometer analyses. (B) Schematic representation of enzymatic cell detachment treatment of the basolateral compartment from IEDC co-cultures to determine DC-mediated HIV-1 transmission via cell-cell contact to U87.CD4.CCR5 cell line. (C-D) IEDC co-culture models were apically exposed to HIV-1 NL4.3BaL for 72h or left non-infected, followed by Accutase-mediated cell detachment treatment to harvest the basolateral DCs. Basolateral DCs were subsequently washed to remove input virus and co-cultured with HIV-1 permissive U87. CD4.CCR5 target cells for 96h. DC marker CD11c was used to exclude single DCs and DC-U87 conjugates from flow cytometry analyses [12]. HIV-1 infection of U87.CD4.CCR5 in co-culture with DCs was thereby defined as CD11c⁻p24⁺cells and is representative of cell-cell HIV-1 transmission by DCs from IEDC co-cultures, determined by multiparameter intracellular p24 staining by flow cytometer analyses. (C) Representative flow cytometry plots and (D) quantification of flow cytometry data. $n = 3$ donors. Ratio paired $t$-test, *$p < 0.05$.
(TIF)

**S4 Fig. TNF-α-induced disruption of intestinal epithelial integrity.** (A,B) TNF-α treatment (400 ng/mL for 96h) compromises epithelial barrier integrity and paracellular permeability in IE cultures as compared to untreated control. (A) Integrity of IE cultures determined by trans-epithelial electrical resistance (TEER) measurements. TEER value ≥200 Ohm.cm2 (dashed line) indicative of a confluent monolayer. $n = 4$ donors, Student's paired $t$-test, *$p < 0.05$. (B) Paracellular permeability of empty inserts, untreated or TNF-α treated IE cultures, determined by 4 kDa FITC-conjugated dextran (FD4) permeation rate of monolayers at 4h post-FD4 addition. Permeability is expressed as FD4 permeation rate: FD4 basolateral$_{t = 4}$(μg)/FD4 apical$_{t = 0}$(μg). Dashed line (0.36) indicates the average of FD4 permeation rate from empty inserts. Data are mean ± SD of $n = 3$ donors, Student's paired $t$-test.
(TIF)

**S5 Fig. Intraepithelial viral sequestration within enriched LBPA-containing vesicles 48h post HIV-1 exposure.** (A) Colocalization of endocytic vesicle markers in IE cultures upon 48h exposure to HIV-1 Gag-iGFP by confocal microscopy analyses. Endocytic vesicle markers (EEA1, LC3, Rab11a, LBPA, red), HIV-1 (green, filled arrowheads). Scale bar = 20 micron, representative of $n = 3$ donors. (B) Pearson's correlation coefficients for colocalization analyses between each endocytic vesicle marker and HIV-1, $n = 3$ donors, Two-way ANOVA, **$p < 0.01$, ***$p < 0.001$.
(TIF)

**S1 Data. Data set.**
(XLSX)

# Acknowledgments

HIS mouse facility (Amsterdam UMC, The Netherlands) is acknowledged for providing fetal tissues. The authors would like to thank Dr. Kees Weijer, Mrs. Esther Siteur-van Rijnstra, Mrs.

Cynthia A van der Linden, and Dr. Arie Voordouw for facilitating the provision and initial processing of the fetal material and Dr. Athanasios Koulis for assisting in organoid culturing. We thank Dr. Ikrame Aknouch and Lance Mulder for sharing protocols for epithelial mono-layer formation and immunostaining, respectively. We thank the Cellular Imaging Core Facility of the Amsterdam UMC for technical assistance during flow cytometry and advanced light microscopy data acquisition. Graphical illustrations in Figs 1A, 2A, 3A, 3F, 7D and S3B were created with BioRender.com (https://biorender.com) under academic subscription.

## Author Contributions

**Conceptualization:** Anusca G. Rader, Dasja Pajkrt, Katja C. Wolthers, Adithya Sridhar, Renée R. C. E. Schreurs, Carla M. S. Ribeiro.

**Data curation:** Anusca G. Rader, Alexandra P. M. Cloherty.

**Formal analysis:** Anusca G. Rader, Carla M. S. Ribeiro.

**Funding acquisition:** Alexandra P. M. Cloherty, Dasja Pajkrt, Katja C. Wolthers, Adithya Sridhar, Renée R. C. E. Schreurs, Carla M. S. Ribeiro.

**Investigation:** Anusca G. Rader, Alexandra P. M. Cloherty, Kharishma S. Patel, Dima D. A. Almandawi, Renée R. C. E. Schreurs, Carla M. S. Ribeiro.

**Methodology:** Anusca G. Rader, Alexandra P. M. Cloherty, Kharishma S. Patel, Dima D. A. Almandawi, Adithya Sridhar, Sterre van Piggelen, Liselotte E. Baaij, Renée R. C. E. Schreurs, Carla M. S. Ribeiro.

**Project administration:** Anusca G. Rader, Carla M. S. Ribeiro.

**Resources:** Anusca G. Rader.

**Supervision:** Carla M. S. Ribeiro.

**Validation:** Anusca G. Rader.

**Visualization:** Anusca G. Rader.

**Writing – original draft:** Anusca G. Rader, Alexandra P. M. Cloherty, Carla M. S. Ribeiro.

**Writing – review & editing:** Anusca G. Rader, Carla M. S. Ribeiro.

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
