## [Decision Letter · Decision Letter 0]

22 Apr 2024

Dear Dr. Ribeiro,

Thank you very much for submitting your manuscript "HIV-1 exploits LBPA-dependent intraepithelial trafficking for productive infection of human intestinal mucosa" for consideration at PLOS Pathogens. As with all papers reviewed by the journal, your manuscript was reviewed by members of the editorial board and by 3 independent reviewers. In light of the reviews (below this email), we would like to invite the resubmission of a significantly-revised version that takes into account the reviewers' comments.

All reviewers found merit in the work, but raised concerns that should be addressed.

We cannot make any decision about publication until we have seen the revised manuscript and your response to the reviewers' comments. Your revised manuscript is also likely to be sent to reviewers for further evaluation.

Sincerely,

Jason M. Brenchley

Academic Editor

PLOS Pathogens

Susan Ross

Section Editor

PLOS Pathogens

Michael Malim

Editor-in-Chief

PLOS Pathogens

orcid.org/0000-0002-7699-2064

All reviewers found merit in the work, but raised concerns that should be addressed

Reviewer's Responses to Questions

**Part I - Summary**

Reviewer #1: The manuscript “HIV-1 exploits LBPA-dependent intraepithelial trafficking for productive infection of human intestinal mucosa” by Rader et al. describes a mechanism by which epithelial cells transmit HIV to underlying target cells. The authors establish a polarized immune-organoid model – consisting of apical intestinal epithelial cells and basolateral monocyte-derived cells - to demonstrate a role for LBPA-rich endocytic vesicles in mediating transcellular transport of HIV across a polarized epithelial barrier. The authors demonstrate that the addition of DCs to the epithelial model amplifies HIV-1 transmission, that pre-treatment of epithelial cells with tenofovir does not reduce their ability to transmit virus, and that treatment of the immune-organoid culture with anti-LBPA reduces transmission. The studies described in this manuscript describe a novel mechanism in which epithelial endosomes contribute to transcellular transport and may inform the design of novel inhibitors of epithelial transcytosis.

The work presented in this manuscript convincingly presents several interesting observations and is clearly representative of significant effort. It is unclear however, whether the focus of this work should be the utility of the IEDC model or the role of LBPA-mediated epithelial transcytosis in seeding/amplifying intestinal infection. The authors describe the results from several experiments that address these topics but do not sufficiently explore either individually. Although the results and insights conveyed by this manuscript are of interest and potential importance, the manuscript does not yet fully support the conclusions drawn by the authors.

Reviewer #2: In this article, Rader et al use an in vitro primary intestinal epithelial culture system to examine the interaction of these cells with HIV-1 and Mo-derived DCs. These interactions have relevance to mucosal HIV-1 transmission and likely can only be studied on a mechanistic level through these in vitro epithelial monolayer systems. There are limitations to this system that are inherent drawbacks for any study employing this model, such as the lack of mucus, commensal microbes, and resident immune cells which are all indispensable components of the gut epithelium. The system is further complicated by the fact that the authors inoculate the culture with HIV at very high titers, which is likely not relevant for early events of transmission. The strength of this article is the author's finding that epithelial uptake of HIV can be blocked in vitro by an LBPA antibody, which to my knowledge is novel. There are certain aspects of these experiments however that within this in vitro model, the authors could have employed designs that are more physiologically relevant. Notably by testing direct DC cell-to-cell transmission to targets rather than the cell-free supernatant used in Figure 2. I also am very surprised at the very high level of p24 staining in DCs. The authors do not provide representative flow dot plots as they show for the HIV permissive cell line (which are surprisingly much lower for p24 than DC). Lastly, there are some statements on extending this to HIV persistence throughout the discussion that should be tempered, as this system is at most a 17 day in vitro culture.

Reviewer #3: The current manuscript by Rader and colleagues titled “HIV-1 exploits LBPA-dependent intraepithelial trafficking for productive infection of human intestinal mucosa" describes a 2D primary intestinal epithelium model, derived from human 3D organoids, for studying mechanisms of enteric viral entry. The authors here show a human intestinal immune-organoid HIV-1 infection using polarized intestinal epithelium and human DCs. The model is useful for studies on viral infection mechanisms in vitro and antiviral drug screening.

Data shows when using multiple donors, incubation with epithelial cell-derived basolateral medium resulted in the productive HIV-1 infection of U87.CD4.CCR5 target cells. Human intestinal epithelium-DC (IEDC) co-cultures were established via sequential seeding of primary intestinal epithelial stem cells followed by primary DCs onto a permeable membrane (3 micron inserts). They found that junctional complex expression patterns at cell-cell borders (i.e., ZO-1), and adherence junctions (e-cadherin) remained unaltered upon HIV-1 inoculation, in contrast to the positive control of TNF-α-induced disruption of intestinal epithelial integrity. Other experiments also concluded that HIV-1 follows a specialized LBPA-dependent transcellular trafficking route within human intestinal epithelium. Overall, I found the paper very novel and potentially significant in generating reagents that other colleagues in the field can also use for detailed viral spread and ART studies in cells other than T-cells.

**Part II – Major Issues: Key Experiments Required for Acceptance**

Reviewer #1: • The authors state that “exposure of intestinal epithelium to HIV-1 results in viral internalization… in a manner that is impervious to antiretroviral treatment”, in the abstract and at other points throughout the manuscript. Although the authors demonstrate that treatment of epithelial cells with tenofovir allows for subsequent virus amplification in U87.CD4.CCR5 cells, the authors utilized only a reverse transcriptase inhibitor. Did the authors use other classes of antiretrovirals - was internalization/transmission still ‘impervious? Could the authors use single-round viruses to further explore the transmission potential of epithelial cells in the context of antiretrovirals? Using a single-round virus, increased transmission potential from antiretroviral pre-treated epithelial cells as compared to treated lamina propria cells or dendritic cells would more convincingly demonstrate the antiviral-resistant transmission potential of epithelial cells.

• The authors describe that the basolateral cells their IEDC co-cultures (Figure 2 and Lines 134-151) are predominantly CD11c+ cells but they do not confirm that these cells are DCs by alternate methods nor do they describe the remaining (CD11c-) cell populations or their contribution to HIV amplification in their model. Although the authors suggest that their methodology produces “high-purity immature DCs expressing canonical DC markers” (Line 340 and Figure S1) independent of adhesion, they do not appear to establish that their post-adhesion, basolateral-associated monocytes are differentiated DCs. Indeed only 75% of their basolateral detached DCs (Figure 2F) are CD11c+, which itself is not an exclusive DC marker. A more detailed phenotypic description of the putative DCs in Figure 2F and of the contribution of the non-DC compartment to transmission in the IEDC model would strengthen the proposed contributions of DCs to viral transmission in the IEDC model.

• Although the IEDC model adds biological relevance, its use in Figure 5F-H muddles the interpretation that LBPA-mediated epithelial transcellular transport serves as an important source of viral transmission during HIV acquisition across an epithelial barrier. This mechanism is addressed more succinctly in Figure S4 – without the addition of DCs - though more replicates are needed to convincingly demonstrate significance. Alternately, did the authors try a step-wise approach in their method development – treating epithelial cells with anti-LBPA and HIV (followed by wash-out), prior to DC attachment?

Reviewer #2: - Figure 1E. It is not clear at what point after in vitro culture of epithelial cells the authors test HIV-1 transmission of U87 cells across the monolayer. This is important because the epithelial layer does not appear to gain integrity until D14 (Fig 1C.D), and what the authors could me measuring may be passive diffusion of HIV.

- Figure 2F. It is peculiar that p24 expression in detached primary CD11c+ DCs is much higher (60% in some samples) than in a CCR5-expressing permissive cell line (Figure 1G, H). The primary DCs are likely not productively infected and I am skeptical that gag would be amplified to a degree that would permit such high p24 detection in the DCs. How was the p24 gate set for figure 1F?? The authors only show CD11c gating. The authors need to show representative flow dot plots of DC p24 expression plus or minus HIV.

- Does HIV activate the detatched dendritic cells by CD80/86 expression as the authors show for LPS in supplemental? This is again a confirmation of the very high p24 expression in DC the authors are showing in Figure 2F.

- Figure 2G, H. It is unclear why the authors chose to culture target cells with IEDC supernatant and not the DC’s themselves given that the authors have demonstrated very high intracellular p24 in the DCs (Figure 2F). The latter would mimic cell-to-cell spread which is likely the more physiologically-relevant scenario.

- Line 164-165. “Together, these findings indicate that apical exposure of intact intestinal epithelium to HIV-1 does not readily compromise tight junction function nor barrier permeability.” The physiological significance of this finding is unclear as the authors are using a system devoid of critical cellular/microbial components of the intestinal epithelium. In vivo, damage to these components (ie, CD4 T cells) by HIV or amplification of pro-inflammatory cytokines by resident innate immune cells most certainly compromises the epithelium.

- Figure 4. A key finding of this work is localization of HIV to LBPA-containing vesicles. Many readers while versed in HIV-1 immunology/pathogenesis may be naïve to the basic biology of endosomal trafficking in epithelial cells. It would help, preferably in the intro or as a preface to Figure 4 to provide background on the underlying biology of all the endosomal markers and their significance.

- Line 228-229: “For example, in both in vitro and in vivo settings, lung bronchial epithelium harbors latently infected cells, which can be reactivated and resume HIV-1 replication upon inflammation.” I would be extremely careful about this statement. The paper the authors are citing (ref 39) only showed vDNA in an in vitro epithelial cell system but to my knowledge no study has shown integrated HIV DNA in epithelial cells of PLWH.

- Line 241-243. “The lasting sequestration of HIV-1 within LBPA-containing vesicles may have additional implications beyond those showcased in the current study, including establishment of latent tissue viral reservoirs and HIV-1 associated immune activation.” The authors again need to be very careful here as their own data only shows sequestration of HIV by epithelial cells for 72 hours and extending this to latent tissue reservoirs in PWH, which persist for years/decades, is overzealous in my opinion.

Reviewer #3: My concerns are:

1. Although the authors have cited few papers on HIV-1 penetration of renal, tonsil, foreskin, and cervical epithelial cells (37, 40), they have not cited some of the pioneering ex-plant work from Leonid Margolis lab (NIH) and some of the recent 3D CNS and HIV infection papers.

2. The virology portion of the paper is rather weak and it could use multiple viral isolates (R5, X4) and some clinical isolates to further validate and also add value to the paper.

3. The cART portion of the paper is also weak and could use titrations and also different combinations of entry and exit inhibitors to at least solidify the receptor mediate entry of the virus.

**Part III – Minor Issues: Editorial and Data Presentation Modifications**

Reviewer #1: • Line 35: The authors state that their findings “demonstrate the pivotal role of intraepithelial multivesicular endosomes as niches for virulent HIV-1”. The authors similarly allude to epithelial cells as a viral reservoir in Lines 265-268. It is this reviewer’s understanding that epithelial cells and endosomes are short-lived - could the authors comment on the longevity of epithelial cells/epithelial-endosomes such that they might contribute to the HIV reservoir?

• Line 47: The authors state that “we have uncovered that HIV-1 specifically hijacks vesicles studded with a particular molecule (LBPA)”. Though this may be true, the authors do not address whether endosomal reorganization within epithelial cells is directly mediated by a viral component (hijacking) versus an anti-viral response by epithelial cells that is indirectly beneficial to the virus.

• Lines 144-146 and Figure 2F - this figure/result is puzzling. The figure legend states “HIV-1 infection of basolateral DCs as a percentage of CD11c+p24+ detached cells, determined by flow cytometer”. Were 40% of CD11c+ cells p24+ or did the authors confirm infection in CD11c+p24+ cells by another manner. Further, 40% “infection” seems abnormally high. Is this true infection or sampling - do the authors have data to confirm infection?

• Figure S2B: the empty insert control (dashed line) for the FD4 permeation rate is far below what is shown for other figures and does not match with what is written within the figure legend.

• Figure 4C-D/Lines 187-190: It is somewhat unclear what is being demonstrated/discussed. What are the distances shown in Figure 4C? Are these a linear distance from a central point or the average at a given radius?

• Figure 5/Lines 194-200: It was unclear from the text and the figure what was being discuss/shown – clarity was only found within the methods.

• Although the authors note that their sorting procedures have historically produced high purity populations, the authors did not comment on the purity of their current preparations. The manuscript would be stronger were they to include this information.

• The authors spend considerable effort describing their 2D IEDC model at the beginning of the paper and in Figure 2 but do not refer to it again until Figure 5. To avoid confusion by the reader, it may be more useful to describe the epithelial-centric findings in the beginning of the manuscript and to develop/utilize the IEDC model toward the end as a biologically relevant model.

Reviewer #2: - Line 240. Minor comment. Pro-viral could easily be interpreted as proviral ‘integrated DNA’, this is not what the authors are referring to here.

Reviewer #3: Although the authors have cited few papers on HIV-1 penetration of renal, tonsil, foreskin, and cervical epithelial cells (37, 40), they have not cited some of the pioneering ex-plant work from Leonid Margolis lab (NIH) and some of the recent 3D CNS and HIV infection papers.

PLOS authors have the option to publish the peer review history of their article (what does this mean?). If published, this will include your full peer review and any attached files.

Reviewer #1: No

Reviewer #2: No

Reviewer #3: No
---

## [Decision Letter · Decision Letter 1]

31 Oct 2024

PPATHOGENS-D-24-00554R1HIV-1 exploits LBPA-dependent intraepithelial trafficking for productive infection of human intestinal mucosaPLOS Pathogens Dear Dr. Ribeiro, Thank you for submitting your manuscript to PLOS Pathogens. The reviewers felt that you have addressed their main concerns. One reviewer asked for minor revisions to the manuscript.. Please submit your revised manuscript within 30 days Dec 30 2024 11:59PM. If you will need more time than this to complete your revisions, please reply to this message or contact the journal office at plospathogens@plos.org. Please include the following items when submitting your revised manuscript:*
A rebuttal letter that responds to each point raised by the editor and reviewer(s). You should upload this letter as a separate file labeled 'Response to Reviewers'. This file does not need to include responses to any formatting updates and technical items listed in the 'Journal Requirements' section below.*
A marked-up copy of your manuscript that highlights changes made to the original version. You should upload this as a separate file labeled 'Revised Manuscript with Track Changes'.*
An unmarked version of your revised paper without tracked changes. You should upload this as a separate file labeled 'Manuscript'. If you would like to make changes to your financial disclosure, competing interests statement, or data availability statement, please make these updates within the submission form at the time of resubmission. Guidelines for resubmitting your figure files are available below the reviewer comments at the end of this letter. We look forward to receiving your revised manuscript. Kind regards, Jason M. BrenchleyAcademic EditorPLOS Pathogens Susan RossSection EditorPLOS Pathogens Michael Malim

Editor-in-Chief

PLOS Pathogens

orcid.org/0000-0002-7699-2064 **Journal Requirements:** **Additional Editor Comments (if provided):** All reviewers have now seen the revised manuscript and are mostly satisfied with the revisions provided.

One reviewer has one remaining minor suggestion that should be addressed.**Reviewers' Comments:** Reviewer's Responses to Questions

**Part I - Summary**

Reviewer #1: (No Response)

Reviewer #2: This manuscript is much improved. I just have a minor critique on the new supplemental 3C.

Reviewer #3: The authors have responded well to my comments.

**Part II – Major Issues: Key Experiments Required for Acceptance**

Reviewer #1: (No Response)

Reviewer #2: none

Reviewer #3: They now show that CCR5-tropic HIV-1 variants (including lab-adapted NL4.3BaL as well as transmitted/founder HIV-1 strains CHO58 and THRO), and to lesser extent CXCR4-tropic HIV- 1 variant NL4.3, are transmitted across IEDC co-cultures.

**Part III – Minor Issues: Editorial and Data Presentation Modifications**

Reviewer #1: (No Response)

Reviewer #2: I would recommend that in the revised supplemental figure 3C for viral cell-cell transmission, the authors denote in the figure the 'cd11c-' population as actually representing the U87.cd4.ccr5 permissive cell line for clarity. To the reviewer it was easily mis-interpreted as the "CD11c-negative non-DC fraction" in revised fig 3H,I, which the authors show to not express any p24.

Reviewer #3: None

PLOS authors have the option to publish the peer review history of their article (what does this mean?). If published, this will include your full peer review and any attached files.

Reviewer #1: No

Reviewer #2: No

Reviewer #3: No

---

## [Editor Report · Decision Letter 2]

1 Nov 2024

Dear Dr. Ribeiro,

We are pleased to inform you that your manuscript 'HIV-1 exploits LBPA-dependent intraepithelial trafficking for productive infection of human intestinal mucosa' has been provisionally accepted for publication in PLOS Pathogens.

Best regards,

Jason M. Brenchley

Academic Editor

PLOS Pathogens

Susan Ross

Section Editor

PLOS Pathogens

Michael Malim

Editor-in-Chief

PLOS Pathogens

orcid.org/0000-0002-7699-2064

The authors have addressed all of the concerns raised by all of the reviewers.
---

## [Editor Report · Acceptance letter]

12 Nov 2024

Dear Dr. Ribeiro,

We are delighted to inform you that your manuscript, "HIV-1 exploits LBPA-dependent intraepithelial trafficking for productive infection of human intestinal mucosa," has been formally accepted for publication in PLOS Pathogens.

Best regards,

Michael Malim

Editor-in-Chief

PLOS Pathogens

orcid.org/0000-0002-7699-2064